# Estrogen receptor activation remodels *TEAD1* gene expression to alleviate hepatic steatosis

Christian Sommerauer [ID][1,12], Carlos J Gallardo-Dodd[1,12], Christina Savva[2], Linnea Hases[3,4], Madeleine Birgersson[3,4], Rajitha Indukuri[3,4], Joanne X Shen [ID][5], Pablo Carravilla[1,6], Keyi Geng[1], Jonas Nørskov Søndergaard [ID][1], Clàudia Ferrer-Aumatell[1], Grégoire Mercier[5], Erdinc Sezgin [ID][6], Marion Korach-André[2], Carl Petersson[7], Hannes Hagström[8,9], Volker M Lauschke [ID][5,10,11], Amena Archer [ID][3,4], Cecilia Williams[3,4] & Claudia Kutter [ID][1✉]

## Abstract

**Sex-based differences in obesity-related hepatic malignancies suggest the protective roles of estrogen. Using a preclinical model, we dissected estrogen receptor (ER) isoform-driven molecular responses in high-fat diet (HFD)-induced liver diseases of male and female mice treated with or without an estrogen agonist by integrating liver multi-omics data. We found that selective ER activation recovers HFD-induced molecular and physiological liver phenotypes. HFD and systemic ER activation altered core liver pathways, beyond lipid metabolism, that are consistent between mice and primates. By including patient cohort data, we uncovered that ER-regulated enhancers govern central regulatory and metabolic genes with clinical significance in metabolic dysfunction-associated steatotic liver disease (MASLD) patients, including the transcription factor *TEAD1*. *TEAD1* expression increased in MASLD patients, and its downregulation by short interfering RNA reduced intracellular lipid content. Subsequent TEAD small molecule inhibition improved steatosis in primary human hepatocyte spheroids by suppressing lipogenic pathways. Thus, TEAD1 emerged as a new therapeutic candidate whose inhibition ameliorates hepatic steatosis.**

**Keywords** MASLD; Estrogen Receptor; Multi-omics; Enhancer–Promoter Interaction; TEAD1
**Subject Categories** Chromatin, Transcription & Genomics; Metabolism; Molecular Biology of Disease

## Introduction

The global obesity epidemic poses a substantial risk for metabolic disorders, including liver diseases (Riazi et al, 2022). Prolonged high-calorie diets, like high-fat diet (HFD), induce hepatic lipid accumulation, resulting in hepatic steatosis, the defining hallmark of metabolic dysfunction-associated steatotic liver disease (MASLD), previously known as nonalcoholic fatty liver disease (NAFLD). Persistent dietary imbalance causes steatohepatitis (MASH/NASH), characterized by hepatocyte death, inflammation, and progressive liver fibrosis, potentially developing into cirrhosis and liver cancer (Søndergaard et al, 2022). MASLD prevalence has risen alongside obesity, currently affecting one-third of adults worldwide (Riazi et al, 2022). Yet, approved medications for MASLD treatment are lacking, highlighting the urgency to identify suitable targets.

MASLD occurrence differs greatly between sexes, with lower prevalence in premenopausal women than in men or postmenopausal women (Clark et al, 2002). The female sex hormone estrogen exerts protective roles in the liver, but the underlying molecular mechanisms remain understudied (Lee et al, 2019). Estrogens bind to nuclear estrogen receptors (ERα and ERβ), acting as transcription factors that activate or repress target genes and signaling cascades by either direct DNA interaction or tethering to other transcription factors (Lee et al, 2019; Palmisano et al, 2017).

Estrogen signaling is crucial in females and males. Endogenous estrogen is produced by enzymatic cholesterol conversion in both sexes. In male mice on a conventional diet, the deficiency of the enzyme aromatase leads to hepatic steatosis (Hewitt et al, 2004), and similarly, liver-specific ERα impairment also induces abnormal liver physiology and liver energy metabolism (Zhu et al, 2014; Qiu et al, 2017). Menopausal hormone therapy in women reduces MASLD prevalence, highlighting that estrogen signaling safeguards hepatic energy metabolism (Clark et al, 2002). Modulating estrogen

[1]Department of Microbiology, Tumor, and Cell Biology, Karolinska Institute, Science for Life Laboratory, Solna, Sweden. [2]Department of Medicine, Integrated Cardio Metabolic Center, Karolinska Institute, Huddinge, Sweden. [3]Department of Protein Science, KTH Royal Institute of Technology, Science for Life Laboratory, Stockholm, Sweden. [4]Department of Biosciences and Nutrition, Karolinska Institute, Huddinge, Sweden. [5]Department of Physiology and Pharmacology, Karolinska Institute, Solna, Sweden. [6]Department of Women's and Children's Health, Karolinska Institute, Science for Life Laboratory, Solna, Sweden. [7]Department of Drug Metabolism and Pharmacokinetics, The Healthcare Business of Merck KGaA, Darmstadt, Germany. [8]Department of Medicine Huddinge, Karolinska Institute, Huddinge, Sweden. [9]Division of Hepatology, Department of Upper GI Diseases, Karolinska University Hospital Huddinge, Huddinge, Sweden. [10]Dr. Margarete Fischer-Bosch Institute of Clinical Pharmacology, Stuttgart, Germany. [11]University of Tübingen, Tübingen, Germany. [12]These authors contributed equally: Christian Sommerauer, Carlos J Gallardo-Dodd. ✉E-mail: claudia.kutter@ki.se

**Glossary**

| | | | |
|---|---|---|---|
| CTCF | CCCTC-binding factor | H&E | hematoxylin and eosin |
| CD | control diet | HFD | high-fat diet |
| CHi-C | promoter-capture Hi-C | H3K27ac | histone 3 lysine 27 acetylation |
| ChIP-seq | chromatin immunoprecipitation followed by sequencing | H3K4me1 | histone 3 lysine 4 monomethylation |
| CoA | coenzyme-A | H3K4me3 | histone 3 lysine 4 trimethylation |
| CPM | counts per million | HSC | hepatic stellate cell |
| DAc | differentially acetylated | KEGG | Kyoto Encyclopedia of Genes and Genomes |
| DEG | differentially expressed gene | MASH | metabolic dysfunction-associated steatohepatitis |
| DIP | 4-(2-(3,5-dimethylisoxazol-4-yl)-1H-indol-3-yl)phenol | MASLD | metabolic dysfunction-associated steatotic liver disease |
| DPN | diarylpropionitrile | NAFLD | nonalcoholic fatty liver disease |
| E2 | 17β-estradiol | NAS | NAFLD activity score |
| ECM | extracellular matrix | NASH | nonalcoholic steatohepatitis |
| ER | estrogen receptor | NES | normalized enrichment score |
| ES-E-G | estrogen-sensitive enhancer-gene pair | PCA | principal component analysis |
| FCCP | carbonyl-cyanide 4-trifluoromethoxy-phenylhydrazone | PPT | pyrazole-triol |
| GEO | gene expression omnibus | TAZ | WW domain containing transcription regulator 1 |
| GO | gene ontology | TPM | transcripts per million |
| GPCR | G-protein-coupled receptor | tSNR | transcriptome-based signal-to-noise ratio |
| GSEA | gene set enrichment analysis | TSS | transcription start site |

levels or ER activity affects hepatic molecular changes and MASLD susceptibility (Besse-Patin et al, 2017). Identifying estrogen-responsive factors and pathways can enhance treatment options for obesity-related liver morbidities while avoiding potential estrogen treatment side effects (Boardman et al, 2015).

In this study, we identified sex-specific molecular signatures that link the hepatoprotective role of ERs to downstream effectors in a diet-induced MASLD mouse model. Utilizing an integrative multi-omics approach, we examined transcriptional and chromatin changes in liver leveraging on single-cell and spatial information. Systemic activation of ER isoforms in mice elucidated their distinct hepatoprotective effects. We found that ER-controlled murine key factors, including *TEAD1*, were similarly altered in MASLD patients. We demonstrated that small molecule-based TEAD inhibition reduced lipid accumulation in an organotypic human liver model by suppressing lipogenesis. Collectively, we identified gene regulatory circuits downstream of ER signaling that control hepatic metabolism and determined that network signature-informed interference can ameliorate liver disease phenotypes.

## Results

### HFD severely changes molecular and physiological parameters in male C57BL/6J mice

To assess diet-induced sexual dimorphism in liver transcriptomes resembling early MASLD stages, we fed 5-week-old female and male C57BL/6J inbred mice a control (CD, 10% fat) or high-fat diet (HFD, 60% fat) for 13 weeks (Fig. 1A). Upon HFD, both sexes gained weight (Hases et al, 2020). Males but not females on HFD developed hepatic steatosis, increased liver weight and circulating glucose levels (Figs. 1B and EV1A–D and Table EV1). These findings confirmed that female mice were more protected from HFD than males.

To investigate the underlying molecular effects, we profiled the transcriptome of livers from male and female mice on CD and HFD ($n = 4$) (Datasets EV1–3). Our principal component analysis (PCA) separated our samples primarily by sex (PC1, 59%) and by diet (PC2, 10%) (Fig. 1C). HFD males exhibited more differentially

expressed genes (DEGs) ($n = 714$) than HFD females ($n = 327$) demonstrating that gene expression in males was more susceptible to HFD than in females, irrespective of genes expressed on the sex chromosomes (Fig. 1D, Diet; Dataset EV1). Only a fraction of genes was commonly deregulated between females and males on HFD, further emphasizing the sex disparity in response to dietary stimuli (Fig. EV1E). We further confirmed these findings by quantifying threshold-independent differences for each comparison (Fig. 1E, Diet). Genes deregulated in both sexes or in HFD males were enriched in biological processes (gene ontology, GO) linked to lipid metabolism, while HFD females exhibited enrichment in circadian rhythm (Fig. EV1E; Dataset EV4).

Taken together, we found that male mice responded stronger to HFD than females and these differences could be traced back to major sex differences in liver transcriptomes.

### Systemic activation of ERα and ERβ mitigates diet-induced gene signatures

Given the resilience of female mice to HFD, we tested the hepatoprotective effects of estrogen in males. After 10 weeks, HFD male mice were injected with agonists that selectively activate ERβ (DPN and DIP) (Harrington et al, 2003; González-Granillo et al, 2019), ERα (PPT) (Harrington et al, 2003) or both (E2) (Harrington et al, 2003) every other day for 3 weeks (Fig. 1A). Liver weight and blood glucose levels did not exhibit significant changes with any estrogenic ligand treatment, and total weight was significantly decreased upon DPN treatment (Fig. EV1A–C) (Hases et al, 2020). All ER agonists reduced steatosis compared to vehicle-treated HFD males (Figs. 1B and EV1D). Our PCA showed that agonist-treated HFD males clustered between HFD and CD males, implying attenuation of HFD-induced alterations (Fig. 1C). DIP had the weakest impact on the transcriptome ($n = 163$ DEGs), whereas DPN ($n = 598$ DEGs), E2 ($n = 510$ DEGs) and PPT ($n = 670$ DEGs) had greater effects (Fig. 1D,E). DPN predominantly downregulated genes, while DIP, E2, and PPT treatments had similar proportions of down- and upregulated genes (Fig. 1D).

We formed the union of DEGs across the five male comparisons (CDm vs. HFDm, HFDm vs. DPN/DIP/E2/PPT, $n = 1477$), which

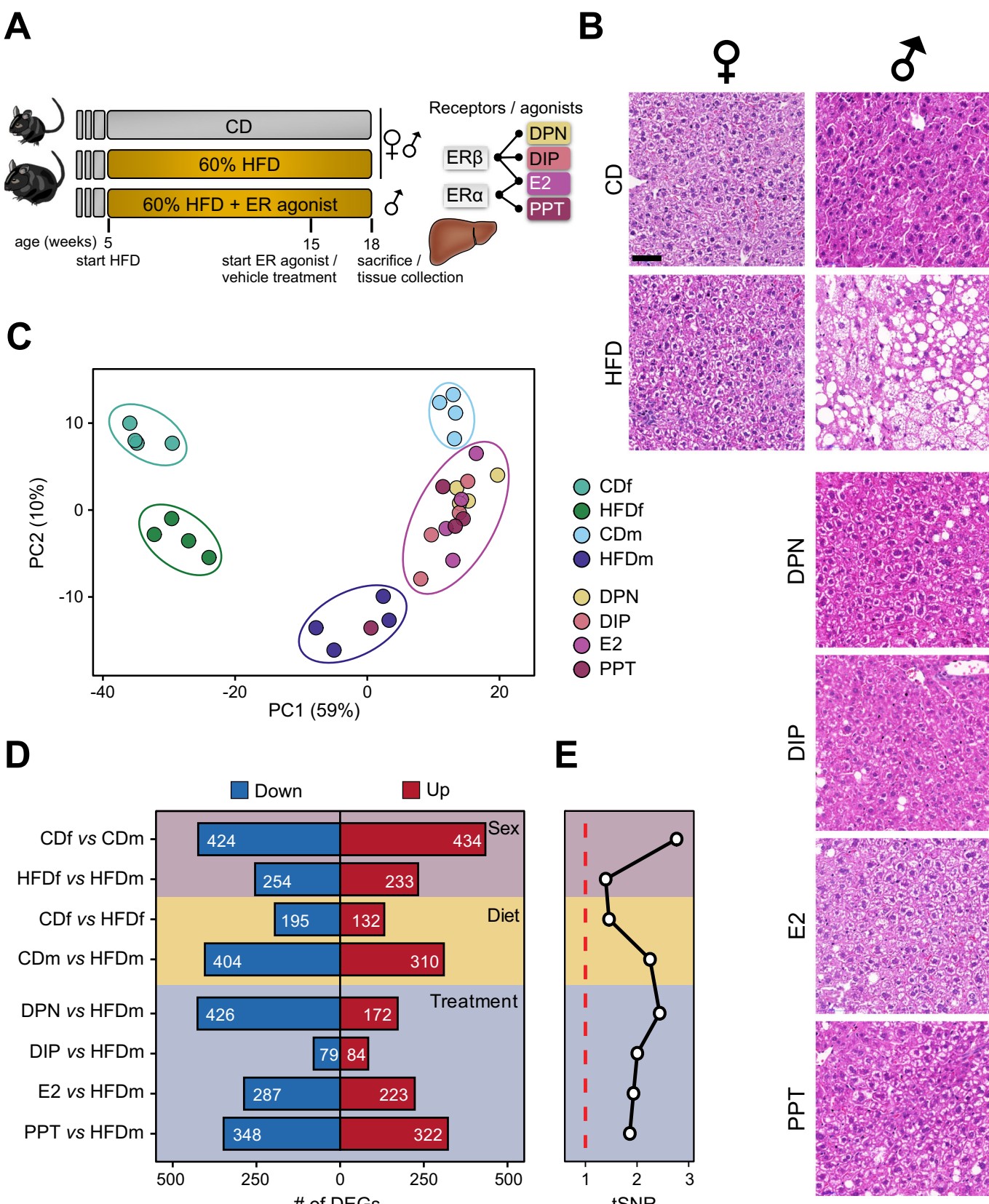

**Figure 1.  Male mice are severely affected by the high-fat diet.**

(**A**) Schematic representation of the mouse experimentation. Five-week-old female (f) and male (m) C57BL/6 mice ($n = 4$) received either a control (CD, 10% fat) or high-fat diet (HFD, 60% fat) for 13 weeks. HFDm subgroups were injected with estrogen receptor α (ERα, E2 or PPT) or ERβ (E2, DPN or DIP) agonists every other day from weeks 15 to 18. Isolated livers were histologically and molecularly assessed. (**B**) Liver cross-sections of female (left) and male (right) mice on different diets and ER-agonist treatments were stained with hematoxylin and eosin. Cross-sections of all four mouse replicates are shown in Fig. EV1D. Scale bar: 50 µm. (**C**) A factorial map of the principal components (PC) analysis separates global gene expression levels. The percentage of PC variance is shown (parentheses). Color-coded small circles illustrate individual mice on different diets and treatments. Color-coded large ellipses group mice by sex, diet, and treatment. (**D**) Horizontal bars present the number (highlighted) of downregulated (blue) and upregulated (red) genes for sex (purple), diet (yellow) or treatment (blue) comparisons ($n = 4$, except PPT: $n = 3$). (**E**) Black line shows the transcriptome-based signal-to-noise ratio (tSNR, $x$ axis) for (**D**) comparisons. Dashed red line represents the noise baseline.

separated into four distinct expression clusters (Fig. 2A). Cluster 1 ($n = 577$) exhibited HFD-induced gene upregulation compared to CD, attenuated by all agonists, while cluster 2 ($n = 258$) displayed HFD-induced gene down regulation, partially restored upon agonist treatment. Cluster 3 ($n = 295$) contained genes with higher HFD expression and ERβ-dependent repression, and cluster 4 ($n = 346$) included genes upregulated by ERα.

We next stratified the DEGs into four categories (Fig. 2B; Dataset EV5). Genes significantly deregulated by HFD were termed "non-reverted" ($n = 335$) when unaffected by ER-agonist treatment and "reverted" ($n = 379$) when restored by at least one treatment. Most of these genes resided in clusters 1 and 2, suggesting an overall adjustment towards the CD state (Fig. 2B). In addition, we distinguished "ERβ-specific" (DPN-DIP, $n = 239$) and "ERα-specific" (E2-PPT, $n = 411$) gene signatures with unchanged expression levels upon HFD but altered upon ER-agonist treatment. Although E2 activates ERα and ERβ, we found a higher overlap between E2- and PPT- than E2- and DPN-regulated genes, indicating that E2 primarily acted through ERα (Fig. EV2A,B). ERβ-specific genes were mostly in cluster 3, while ERα-specific genes were predominantly in clusters 1 and 4 (Fig. 2B). The degree of recovery varied among ER-agonist treatments, with PPT and DPN showing the highest number of reversed HFD-deregulated genes (38% and 37%, respectively), followed by E2 (35%) and DIP (16%) (Fig. EV2C).

For each of the four categories, we investigated gene enrichments in GO biological processes. The reverted and ERβ-specific gene sets showed significant enrichments of genes regulating lipid metabolism, ERK signaling, xenobiotic metabolism and immune responses (Figs. 2C and EV2D; Dataset EV4). In addition, the ERβ-specific gene sets controlled extracellular matrix organization, apoptosis, cell motility and differentiation processes, which were almost entirely represented in cluster 3 characterized by ERβ-agonist treatment-specific gene downregulation (Figs. 2B and EV2D). We found no GO term overrepresentation for non-reverted and ERα-specific gene categories.

Overall, systemic ERα or ERβ activation restored diet-induced gene expression changes, with isoform-specific differences, correcting metabolic processes to reduce steatotic phenotypes (Fig. 1B).

## Systemic ER activation has widespread implications in core liver pathways

We performed a threshold-independent gene set enrichment analysis (GSEA) to capture functionally relevant genes recovered upon ER-agonist treatments but without reaching statistical significance (Fig. 2). Reactome pathway analysis, clustering, and subsequent correlation based on normalized enrichment scores (NES) identified 24 relevant pathway clusters that were significantly altered in HFD males

compared to CD males and HFD ER-agonist-treated males (Fig. 3A; Appendix Figs. S1 and S2; Dataset EV6).

Upon connecting the pathway clusters, we uncovered that most reverted genes were linked to lipid metabolism (Node N8) and biological oxidations (N21) (Appendix Fig. S2). These genes had lower expression levels in CD males and ER-agonist-treated males compared to HFD males, consistent with our previous findings (Figs. 2 and EV2). While most pathways showed similar effects with both ERα and ERβ activation, we noticed that lipid metabolism was slightly more changed by ERα. Within lipid metabolism, ERα particularly modulated fatty acyl-coenzyme A biosynthesis processes (Appendix Fig. S1). We also uncovered ERβ-dominant effects in regulating phagocytosis (N4), extracellular matrix (ECM, N5), carbohydrate metabolism (N6) and G protein-coupled receptor signaling (N7) (Fig. 3A). ERβ agonists specifically suppressed insulin-like growth factor regulation and ECM-related processes such as collagen formation (Appendix Figs. S1 and S2).

Altogether, our analysis revealed extensive implications of systemic ER activation on central processes beyond lipid metabolism and enabled us to distinguish between shared and ER isoform-specific regulation.

## HFD and ER activation signatures co-occur in the liver across species

Physiological functions of the liver rely on coordinated actions between different cell types. To determine which cell types were affected by HFD and recovered upon the ER-agonist treatments, we analyzed public single-cell (comprising 483,955 cells) and spatial transcriptomics datasets (Guilliams et al, 2022).

After filtering for males, we focused on cells representing 16 annotated cell types (Fig. EV3A). HFD led to a reduction of major liver cell types, including hepatocytes, endothelial and Kupffer cells, while immune cell populations increased (Fig. EV3B). This confirms previous findings and partly explains HFD-induced gene expression changes in the liver (Guilliams et al, 2022; Kovats, 2015). By examining cell type-specific gene expression patterns of our HFD and ER-agonist treatment-derived signatures, we found that the non-reverted, reverted and ERα-specific gene sets (Fig. 2B) were mainly enriched in hepatocytes. In contrast, ERβ-specific effects were prominent in endothelial and stromal cell populations, aligning with the profound effects of ERβ on ECM-related genes, including many collagen genes (Fig. 3B). These observations potentially reflect different ERα and ERβ activities in hepatic cell populations (Karlsson et al, 2021). The same cell types were enriched when mapping these gene signatures to reference human and healthy macaque single-cell liver atlases. This suggested that

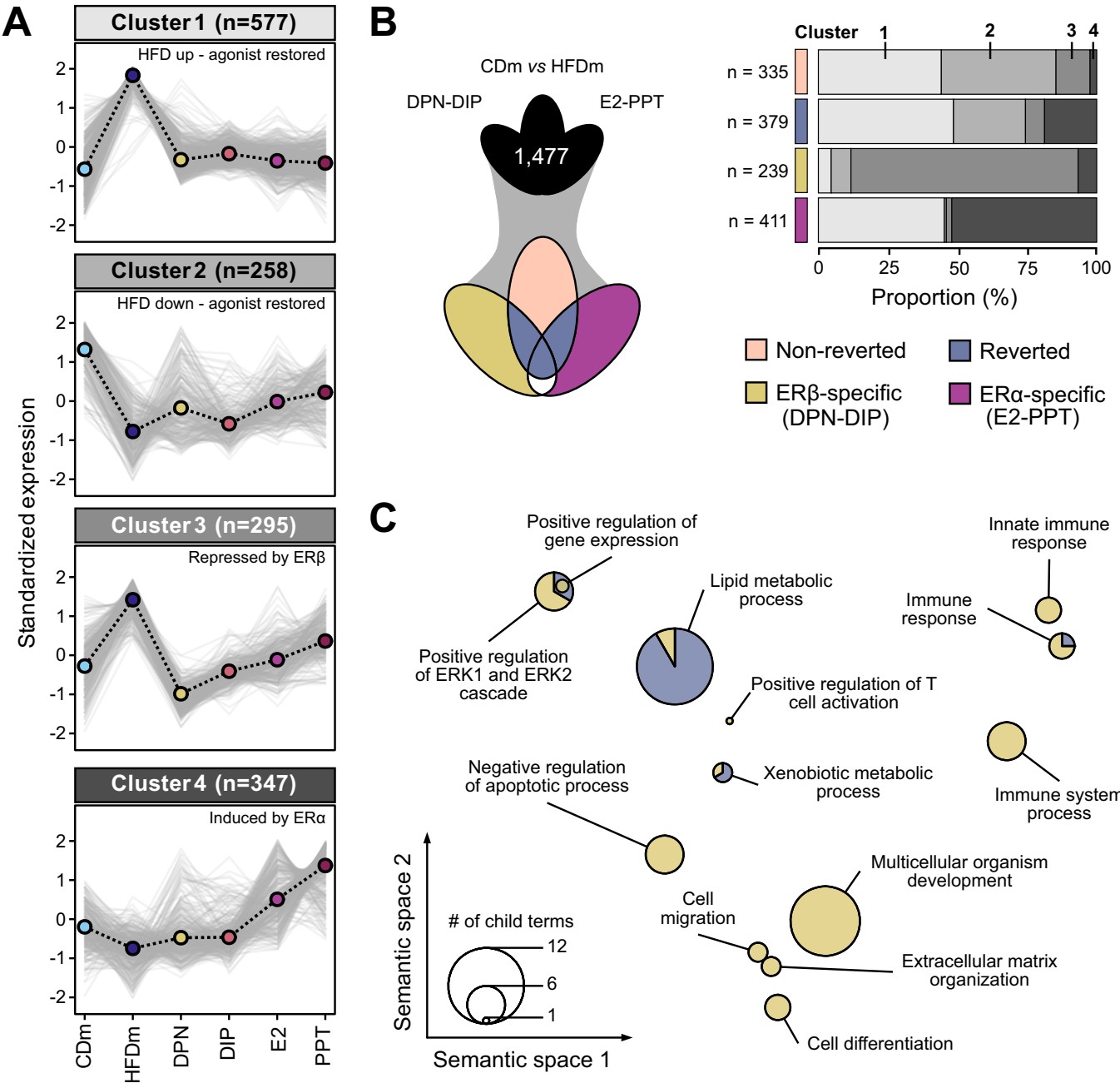

**Figure 2. ERα/β-agonist treatment largely reverts HFD-induced transcriptome alterations in males.**

(A) Line charts depict four clusters (gray-scaled) of gene expression trends (*z*-score) for unified deregulated genes (DEGs, $n = 1477$) in mice on different diets and ER-agonist treatments (color-coded, $n = 4$, except PPT: $n = 3$). Cluster centroid (dashed black line) represents all deregulated genes (gray lines). Number of genes per cluster is shown (parentheses). (B) Three-way Venn diagram (left) presents intersections of unified DEGs (clustered in (A)). Genes are categorized (right) into non-reverted (rose), reverted (denim), ERβ-specific (DPN-DIP, ocher) and ERα-specific (E2-PPT, violet) gene sets. Horizontal bar chart displays the proportional occurrences of gene sets in the four clusters (gray scale as in (A)). Number (*n*) indicates gene set size. (C) PCA factorial map separates the semantic space of enriched gene ontology (GO) terms in reverted and ERβ-specific gene sets (circles). Enriched GO terms are collapsed at the parent term level and separated based on similarity (*x*–*y* axis). Circle size corresponds to number of enriched GO terms. A hypergeometric test with Benjamini–Hochberg correction was used for the overrepresentation analysis.

the hepatic molecular key signatures and the cellular architecture altered by HFD or in MASLD affect similar cell types in mice and humans, and that the observed gene regulatory responses to estrogen treatment are partly shared (Fig. EV3C).

Analyzing spatial transcriptomics data allowed to identify zonation-specific expression patterns of these signatures across the liver lobule. We found that HFD-induced changes were concentrated near the central vein area, while ERβ-specific effects

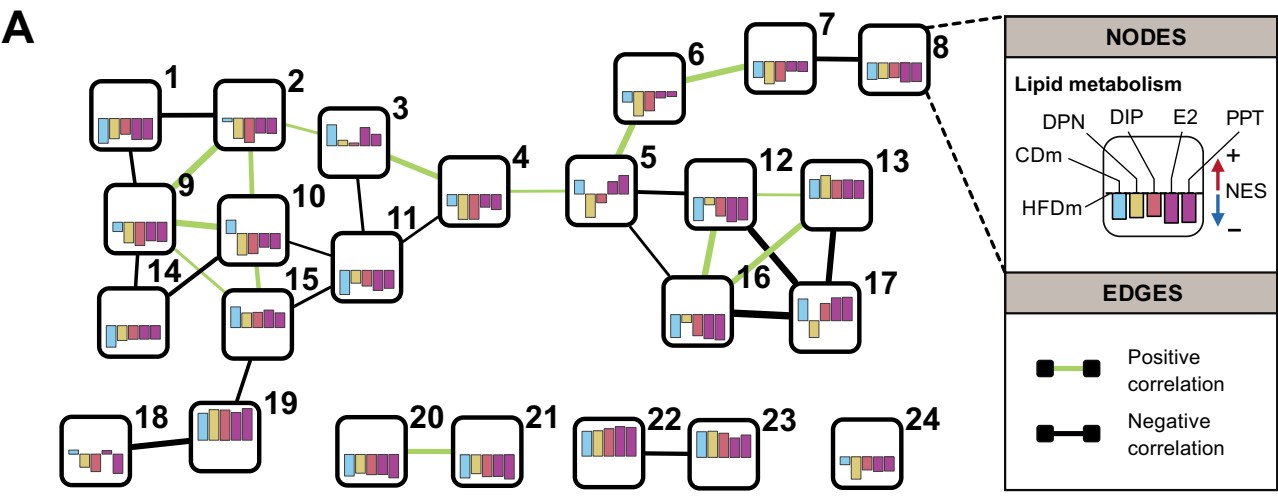

◄ **Figure 3. ERα/β-agonist treatment reverts HFD-induced transcriptional changes by affecting central cellular pathways and liver cell types.**

(A) Network connects major Reactome pathway clusters (numbered nodes). Colored bars inside each node present pathway cluster enrichment for CDm and ER-agonist-treated HFDm compared to HFDm (normalized enrichment score, NES). Edges connect positively-correlated (green) or negatively-correlated (black) nodes based on NES profiles. (B) UMAP space-projected enrichment plots highlight liver cell types with enhanced signal for the gene sets (defined in Fig. 2B). Hepatocyte nuclear fraction (snRNA-seq) is labeled. (C) Spatial transcriptomics maps show liver zonation patterns of the gene sets (defined in Fig. 2B). (D) Bubble plots display activity scores of altered pathway clusters (A) across all liver cell types in control (left), HFD (middle) male mice and their differences (right). Color-code and circle diameter for enrichment score: low (yellow, narrow), high (black, wide). Arrows and enrichment change indicate higher abundance in control (red) and HFD (blue) mice.

were enriched in the vasculature including capsule, portal, and central vein (Fig. 3C).

To characterize the biological roles of individual cell types in the liver, we assessed the enrichment of previously altered pathways (Fig. 3A). We observed that metabolic and oxidative processes occurred in pericentrally located hepatocytes, while processes related to extracellular matrix remodeling operated in stromal cells and the vasculature (Figs. 3D and EV3D,E). In addition, comparing pathway enrichment scores from the control to the HFD condition revealed gene expression changes in immune cells promoting phagocytosis and complement cascade processes (Fig. 3D).

Overall, our findings highlight that HFD primarily perturbed hepatocyte homeostasis by altering crucial metabolic and oxidative processes, leading to mobilization and activation of immune cells. We find that these gene signatures are in part shared between mouse and human, and that systemic ER activation protects the liver by counteracting these changes.

## Activation of ER-responsive pathways is mediated through chromatin changes

The epigenomic and transcriptomic landscapes are intricately linked to maintain cellular homeostasis and can be altered by dietary changes (Siersbaek et al, 2017). To investigate ER-agonist-dependent epigenomic restoration of physiological and transcriptional profiles, we performed chromatin immunoprecipitation followed by sequencing (ChIP-seq) on livers of CD, untreated as well as ER-agonist-treated HFD male mice. We focused on modified histones associated with accessible chromatin at promoters (histone 3 lysine 27 acetylation, H3K27ac and H3K4 trimethylation, H3K4me3) and enhancers (H3K27ac and H3K4 monomethylation, H3K4me1). We identified 12,598 promoters and 26,210 enhancers, of which 142 promoters and 2181 enhancers were differentially acetylated (DAc) at H3K27 upon HFD (Fig. EV4A,B). Most enhancer sites gained H3K27ac in response to HFD (69%), while promoter sites equally gained and lost H3K27ac (Fig. 4A). We found that H3K27ac at both promoters and enhancers were partly restored by all ER agonists (Figs. 4A and EV4C).

Enhancer–promoter interactions through chromatin loops impact gene transcription (Zuin et al, 2022), therefore we examined the involvement of DAc enhancers in regulating nearby HFD-affected genes. Overall, we identified 6543 differentially regulated enhancer-gene (E-G) pairs, of which 80 were estrogen-sensitive with 49 unique paired genes residing within chromatin loops (Fig. EV4D; Appendix Fig. S3; Dataset EV7). These 49 genes were significantly enriched in metabolic processes (Fig. EV4E; Dataset EV4), aligning with the observed transcriptomic changes (Fig. EV2).

Among the estrogen-sensitive enhancer-gene pairs (ES-E-Gs), four enhancers near the *TEA domain transcription factor 1* (*Tead1*) gene showed HFD-induced gain of H3K27ac, which was reduced upon estrogenic ligand treatment (Fig. 4B). Using promoter-capture Hi-C (CHi-C) data, we discovered interactions between *Tead1* and nearby HFD-regulated gene loci through enhancers and chromatin loop formation (Fig. 4B). In addition, we found enhancers across the Acyl-CoA thioesterase (*Acot*) gene loci that were topologically connected via chromatin loops involving the HEAT Repeat Containing 4 (*Heatr4*) gene locus (Fig. EV4F), suggesting a shared regulatory module for several *Acot* genes.

Combined, these results showed that HFD induces major epigenomic rearrangements in livers of male mice and identified 80 ES-E-Gs that provide insights into the regulatory mechanisms involved. Importantly, these alterations are reversible by ER activation, providing a promising basis for therapeutic interventions.

## Expression trends of ES-E-G genes follow MASLD disease progression in humans

Recent liver cohort studies were designed to identify potential biomarkers and drivers of MASLD. We reanalyzed a large MASLD cohort dataset (Govaere et al, 2020) ($n = 216$) to examine the expression levels of ER-reverted orthologs (45/49 genes) in MASLD patients separated by disease severity (CTRL, MASL, and MASH, collectively termed MASLD stages) (Fig. 5A and Table EV2). By applying k-means clustering ($k = 4$) to the gene expression profiles, we identified gene sets that were upregulated (cluster 1, $n = 18$) or downregulated (cluster 2, $n = 11$) with disease progression, as well as genes primarily induced in MASH (cluster 3, $n = 4$) or MASL (cluster 4, $n = 12$) (Fig. 5A, panel 1). These gene expression patterns correlated with the NAFLD activity score (NAS) spectrum (Fig. 5A, panel 2) and exhibited limited consistency across fibrosis stages (Fig. 5A, panel 3). Notably, there was strong consistency in the directionality of ER-regulated gene expression changes between MASLD patients and HFD male mice (Fig. 5A, panel 4). In contrast, most genes were unaltered in female mice upon HFD (34/45 genes) (Fig. 5A, panel 5).

*TEAD1* gene expression was increased in MASLD patients and HFD male mice (Fig. 5A–C). Unlike the other three gene family members, *TEAD1* is broadly expressed in the liver (Dataset EV8). *TEAD1* encodes a key transcriptional effector of the Hippo pathway, and this pathway has been recently described to regulate liver homeostasis and metabolism (Ardestani et al, 2018; Koo and Guan, 2018). ER-agonist treatment in HFD male mice decreased *Tead1* gene expression (Fig. 5C; Dataset EV8). siRNA-mediated knockdown of *Tead1/TEAD1* reduced lipid droplets and oxygen consumption rates in cell lines, suggesting changes in energy metabolism (Figs. 5D,E and EV5A; Table EV3; Dataset EV9). In a physiologically relevant human model, we treated primary human

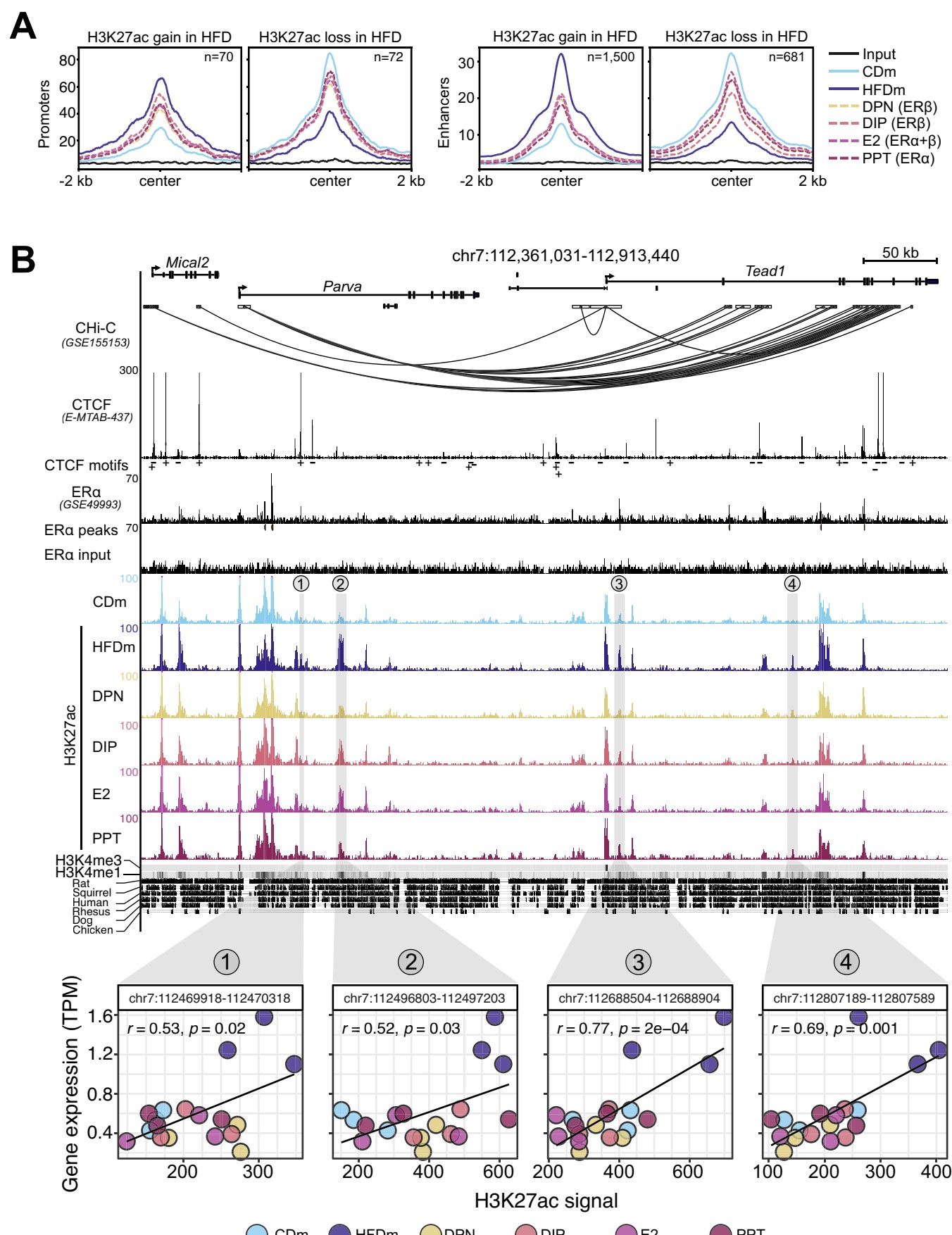

**Figure 4. ERα/β-agonist treatment recovers HFD-induced changes at enhancers and promoters.**

(A) Metaplots show H3K27ac read aggregation in promoters (left) and enhancers (right), centered at the peak summits. Number (*n*) indicates significant H3K27ac signal gains or losses in livers of HFDm compared to CDm. The average signal is depicted (*n* = 3 mice per condition). (B) Genome browser view (mm10) illustrates genomic region around the *Tead1* gene locus. Black boxes represent exons and UTRs. Arrows indicate gene transcription directionality. The scale bar shows genomic region length in kilobases (kb). Black arcs display promoter-capture Hi-C (CHi-C) 3D connections. Genomic regions are enriched for CTCF (black peaks) with CTCF motif orientations determined with FIMO (plus or minus symbols), ERα (black peaks), ERα input (black peaks), significant ERα peaks (black insets), H3K27ac (color-coded peaks) in CDm, HFDm and ER-agonist-treated HFDm, H3K4me3, and H3K4me1 (horizontal gray bar; dark: high, light: low). One replicate per condition is shown. The *y* axis of each track specifies normalized read density. Genomic location of enhancers (numbered from 1 to 4) paired with the *Tead1* gene locus are highlighted (gray vertical boxes). The degree of genomic sequence conservation in vertebrates is shown (conserved: black, not conserved: white). Scatter plots correlate *Tead1* gene expression (TPM, *y* axis) and its paired enhancers (H3K27ac signal, *x* axis) in the livers of male mice on different diets and ER-agonist treatments. All three biological replicates are shown. Enhancer coordinates (400 bp window around the enhancer summit), Pearson correlation coefficients (*r*) and significance (*p*) are indicated in each box.

hepatocyte (PHH) spheroid cultures (Bell et al, 2016) in steatogenic media with TEADap (VT-104), an inhibitor of TEAD autopalmitoylation disrupting the interaction between TEAD and its cofactor YAP (Tang et al, 2021) as well as TEADsf (Ex.174), a small molecule inhibitor binding directly to the TEAD surface blocking the YAP/TEAD interface (Bordas et al, 2021) (Dataset EV8). Notably, we observed a significant reduction in lipid accumulation with TEADap, exhibiting stronger effects than TEADsf (Fig. 5F).

Overall, we identified networks of ER-controlled genes that overlapped between mouse and human livers and were predictive of MASLD and fibrosis stages. Among the ER target genes that showed similar responses was *Tead1/TEAD1*. In an organotypic human liver model, TEAD inhibition reduced hepatic steatosis.

### Hepatic TEAD inhibition ameliorates steatosis by altering central metabolic pathways

To investigate the molecular changes underlying the reduction of hepatic steatosis by TEAD inhibition, we determined gene expression changes in PHH spheroids treated with the TEAD inhibitors in steatogenic media (Dataset EV10). The TEADap inhibitor induced more DEGs (*n* = 435) compared to the TEADsf inhibitor (*n* = 175), with 125 DEGs shared between both treatments (Fig. 6A). This indicated that both compounds affected similar genes, albeit to different degrees. DEG analysis revealed a large set of repressed genes (cluster 1, *n* = 391) and a smaller set of activated genes (cluster 2, *n* = 94) (Fig. 6B). Pathway analysis (KEGG) of TEADap deregulated genes revealed alterations in molecular metabolism, including AMP-activated protein kinase (AMPK) and phosphatidylinositol-3-kinase (PI3K)-AKT signaling (Figs. 6C and EV5B,C; Dataset EV4), overall resembling a starvation response. TEAD inhibition may disrupt the direct binding of TEAD proteins to promoters of metabolic genes, for example, *SREBF1* (de novo lipogenesis), *HMGCR* (cholesterol synthesis), or *GHR* (growth hormone receptor), and thereby alter cellular energy and lipid homeostasis (Figs. 6C and EV5D).

We then assessed the impact of TEAD on gene regulation through chromatin interactions by quantifying TEAD1-binding sites in DEGs. TEADap DEGs had significantly more TEAD1-binding sites (mean = 0.4 per gene) compared to random size-matched gene sets (mean = 0.17 per gene, range: 0.12–0.25) (Fig. 6D), suggesting direct regulation by TEAD1 rather than secondary signaling mechanisms.

Lastly, we evaluated the contributions of TEAD inhibition to the observed beneficial effects upon ER-agonist treatment. After ortholog conversion, we found that 27.4% (17/62) of significantly enriched genes in the top KEGG pathways after TEADap treatment

of PHH spheroids were also differentially expressed upon ER-agonist treatment in male mice on HFD, compared to random size-matched gene sets (median: 9.7%) (Fig. EV5E). Moreover, the gene expression trends in ER-agonist-treated HFD male mice closely resembled those in TEADap-treated PHH spheroids (Fig. EV5C), suggesting that ER-agonist treatment partially restored MASLD in a TEAD-dependent manner.

To summarize, we demonstrated that systemic estrogen signaling suppresses *Tead1* gene expression in HFD male mice and inhibition of TEAD reduces lipid accumulation in human hepatocytes by repressing crucial lipogenic pathways (Fig. 6E).

## Discussion

Beyond reproductive roles, estrogen signaling also maintains tissue homeostasis and estrogenic benefits are well recognized in postmenopausal women and men (Hammes and Levin, 2019; Clark et al, 2002). Similarly, estrogen treatment in male mice alleviated metabolic syndrome, including steatosis and insulin resistance (Wang et al, 2015). Our study revealed that estrogenic agonist treatment restored deregulated lipid metabolism and oxidative processes, highlighting the positive metabolic effects of ER activation. Furthermore, we uncovered previously overlooked cellular pathways affected by estrogen signaling, emphasizing its role in maintaining liver homeostasis besides lipid metabolism.

Previous studies involving Estrogen receptor alpha (*Esr1*) gene deletions in both sexes have established ERα as a hepatic key regulator of lipid metabolism, gluconeogenesis and other essential metabolic processes (Palmisano et al, 2017; Lee et al, 2019). Various dietary disease models have confirmed the protective role of ERα in MASLD. However, the functional impact of hepatic ERα in safeguarding the liver upon dietary stress remains ambiguous, reporting its requirement (Qiu et al, 2017; Zhu et al, 2014; Meda et al, 2020; Wang et al, 2015) and dispensability (Matic et al, 2013; Hart-Unger et al, 2017; Meda et al, 2020). This ambiguity may stem from developmental shifts in metabolic regulation that affect adult liver function. In our study, ER-agonist treatments in adult mice eliminated congenital confounders and revealed that ERβ activation overall mirrors the cellular and molecular phenotypes observed for ERα signaling (Yepuru et al, 2010). While ERβ is not expressed in hepatocytes (Dataset EV11), it likely regulates hepatic metabolism through other cell types, such as immune cells present in the liver (Kovats, 2015). Estrogens possess anti-inflammatory properties (Straub, 2007), suggesting that ERα and ERβ contribute to liver homeostasis through immune cells or systemic anti-inflammatory signaling pathways. The analysis of single-cell data confirmed that

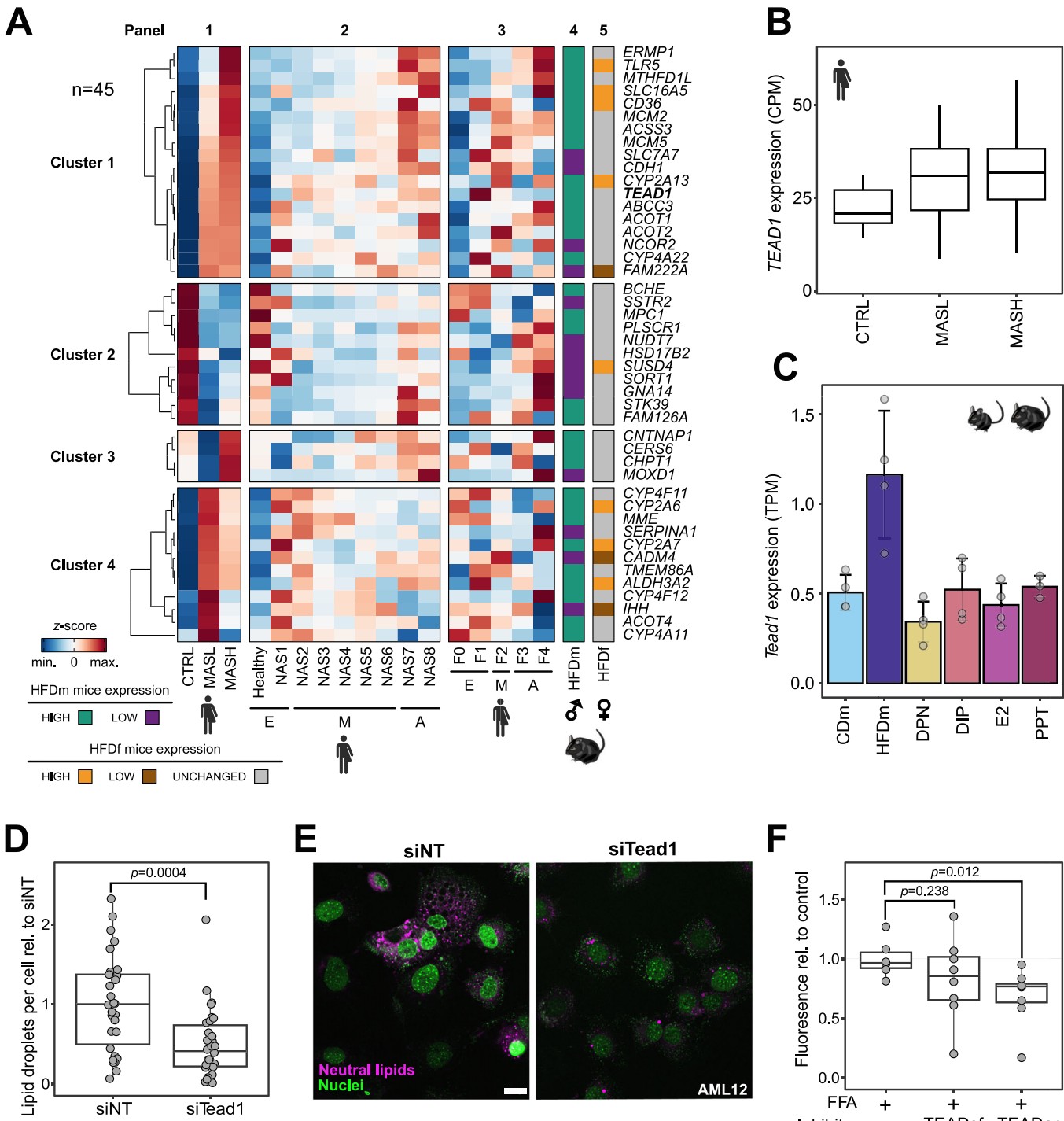

**A** Panel

z-score
min. 0 max.

HFDm mice expression
HIGH    LOW

HFDf mice expression
HIGH    LOW    UNCHANGED

**B**

*TEAD1* expression (CPM)

CTRL    MASL    MASH

**C**

*Tead1* expression (TPM)

CDm    HFDm    DPN    DIP    E2    PPT

**D**

Lipid droplets per cell rel. to siNT

*p*=0.0004

siNT    siTead1

**E**

siNT    siTead1

Neutral lipids
Nuclei    AML12

**F**

Fluorescence rel. to control

*p*=0.012
*p*=0.238

FFA        +    +    +
Inhibitor  −    TEADsf    TEADap

HFD induces inflammatory signaling and alters hepatic immune cell composition, potentially amplifying the responsiveness to or effects by estrogens due to increased proportions of immune cells (Kovats, 2015). Notably, ER-agonist treatments restored the expression of genes involved in monocyte recruitment and inflammatory signaling in HFD male mice.

Low ERβ gene expression was detected in hepatic stellate cells (HSCs) (Karlsson et al, 2021), which contribute to fibrosis upon activation. ERβ could mitigate HSC activation and attenuate liver fibrosis. While the expression of fibrosis-associated genes was generally unchanged in our HFD model, the ERβ agonists specifically and predominantly suppressed a range of genes associated with the extracellular matrix, angiogenesis and growth factor signaling. Many of these genes are known to be markedly upregulated upon HSC activation during fibrosis, including Collagen Type I and Type III Alpha 1 Chain (*Col1a1* and *Col3a1*) (Bourd-Boittin et al, 2011). Treatment with ERβ agonists may pose a future treatment strategy for diet-induced fibrosis (Zhang et al,

◀ **Figure 5.   ER-sensitive genes are associated with MASLD progression and reveal TEAD1 as a clinical target.**

(A) Heatmap displays changes in expression levels for the 45 orthologous ES-E-G genes in MASLD patients (panels 1–3, $n = 216$) and mice (panels 4 and 5, $n = 4$). Color gradient indicates $z$-score-normalized gene expression counts (blue: low, red: high). Four $k$-means clusters group genes by expression in healthy (CTRL), MASL and MASH patients (panel 1) as well as patients with different NAS (early (E): NAS0-1, moderate (M): NAS2-6, advanced (A): NAS7-8, panel 2) and fibrosis stages (E: F0-1, M: F2, A: F3-4, panel 3). Expression levels of the 45 genes in HFDm (panel 4) and HFDf (panel 5) mice are shown. Color codes distinguish downregulated and upregulated genes in HFDm *versus* CDm (purple: low, green: high) and HFDf *versus* CDf (brown: low, gray: unchanged, orange: high). Gene names follow human nomenclature. (B) Box plot shows CPM-normalized *TEAD1* gene expression in the MASLD patient cohort depicted in (A). Each box indicates the interquartile range (IQR), median (horizontal line) and 1.5×IQR (whiskers). (C) Bar chart displays TPM-normalized *Tead1* gene expression in male mice ($n = 4$ per condition, ±SD). Color gradient indicates male mice on different diets (CD or HFD) and upon HFD and ERα/β-agonist treatments (DPN, DIP, E2 or PPT). Dots indicate individual mice. (D) Box plot shows microscopically quantified lipid droplet number in AML12 cells with siRNA-mediated *Tead1* knockdown (siTead1) relative to control (siNT) ($n = 2$ biological and $n = 5$ technical replicates). Each box indicates the interquartile range (IQR), median (horizontal line) and 1.5×IQR (whiskers). Dots indicate individual images. $P$ value is shown (two-sided $t$ test). (E) Representative images of AML12 cells transfected with siRNA nontargeting (siNT, left) and siRNA targeting *Tead1* (siTead1, right). Neutral lipids (LipidTox, purple) and nuclei (NucBlue, green) were stained. Scale bar: 20 µm. (F) Box plot depicts fluorescently measured lipid content in free fatty acid-fed (FFA + ) human primary hepatocyte spheroids ($n = 22$). Dots indicate individual spheroids. $P$ values are shown (two-sided $t$ test). Each box indicates the interquartile range (IQR), median (horizontal line) and 1.5×IQR (whiskers).

2018). The transmembrane-bound G protein-coupled ER (GPER1), which can be activated by E2 and PPT but not by DPN, may partially mediate the effects observed with E2 and PPT (Palmisano et al, 2017). Although GPER1 expression was undetectable in our mouse liver data, previous reports demonstrated that GPER1 deficiency in male mice leads to dyslipidemia (Sharma et al, 2013). Future research involving cell type-specific deletions of ERα, ERβ and GPER1 will be needed to dissect the crosstalk between different cell populations and tissues.

Our study revealed that systemic ERα and ERβ activation reversed HFD-induced alterations in enhancer and promoter accessibility. Identifying enhancers is clinically relevant since they can be targeted therapeutically by interfering enhancer RNA (Huang et al, 2021; Sommerauer and Kutter, 2022). We determined a stringent set of 80 ES-E-Gs, including genes associated with MASLD. Notably, we discovered a large enhancer locus near the *Acot* genes with multiple ERα binding sites, suggesting direct regulation by ERα in the liver. These genes regulate β-oxidation through peroxisome proliferator-activated receptor alpha (PPARα) (Franklin et al, 2017), thus linking ERα activation to lipid catabolism. Furthermore, we found four enhancers near the *Tead1* gene locus, which exhibited increased activity by HFD and restoration upon estrogenic ligand treatment. While the direct regulation of the *TEAD1* gene by ERs in hepatocytes is plausible, our data cannot preclude the involvement of various hepatic cell types and secondary signals, which requires further exploration. In MASLD, the Hippo pathway co-factors YAP and TAZ have recently been investigated (Ardestani et al, 2018; Koo and Guan, 2018), however, the roles and regulation of TEAD1 has been largely unexplored, mainly due to embryonic lethality upon knockout (Chen et al, 1994). Moreover, the involvement of the Hippo pathway in energy metabolism in cancer cells promoted the development of drugs targeting oncogenic TEAD which could be repurposed for MASLD (Pobbati et al, 2023).

Many MASLD treatments targeting metabolic regulators showed efficacy in mice but failed in clinical trials (Xiao et al, 2021). Therefore, our study focused on identifying genes regulated similarly in mouse and human. Most of our mouse gene candidates exhibited consistent gene expression trends in human, suggesting translatable responsiveness to estrogenic ligand treatment. Specifically, *TEAD1* exhibited similar gene expression trends in HFD male mice and MASLD patients. The Hippo pathway, known for regulating tissue homeostasis, is implicated in metabolic disease (Ardestani et al, 2018). Our findings support the notion that Hippo signaling, through TEAD deregulation, activates catabolic metabolic pathways, including cholesterol and fatty

acid synthesis upon energy surplus. In conclusion, inhibiting TEAD and its interaction with YAP presents a promising new therapeutic strategy for metabolic diseases, like MASLD, bypassing potential adverse effects of estrogen treatment.

## Limitations

This study primarily examined the effect of ER-agonist treatment on male mice, given that female mice fed with a high-fat diet showed protection against hepatic steatosis. Although our findings indicate that the ER-sensitive genes identified in males are also controlled by estrogen signaling in female mice, further assessment is required to determine whether these genes partially account for the sex disparity observed in MASLD. Moreover, our study investigated only a single time point after 3 weeks of ER-agonist treatment. Consequently, the identified ER-sensitive genes are likely a combination of direct and indirect effects of ER signaling. The detection of gene signatures relied on transcriptomic differences from bulk liver samples, potentially overlooking subtle changes in low-abundant cell types. Furthermore, inter-individual variability, including demographic, environmental and genetic factors can impact outcomes when working with human primary cells. However, previously we did not observe major differences in the molecular effects of YAP/TEAD inhibitors (Oliva-Vilarnau et al, 2023), despite MASH phenotypes slightly varied when growing spheroids derived from different donors. While these findings argue against major differences of the molecular networks underlying YAP/TEAD inhibition, a modulating role of various individual factors cannot be excluded.

## Methods

### Ethics approval and consent to participate

All experimental protocols were approved (N230/15) by the local ethical committee of the Swedish National Board of Animal Research.

### Animal experiments and tissue preparation

Animal experimentation has been previously reported and ARRIVE guidelines were followed (Hases et al, 2020). In short, five- to

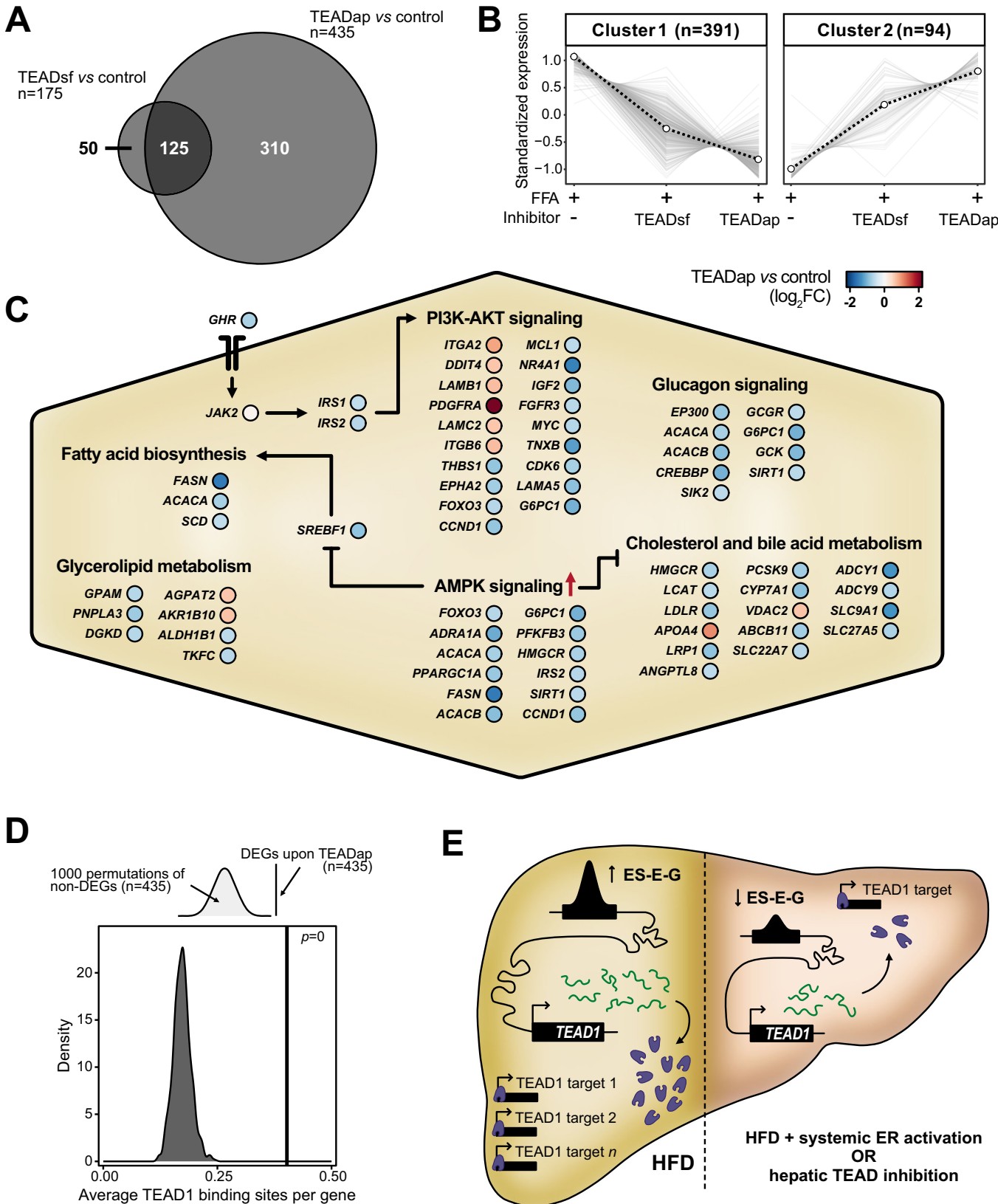

**Figure 6. TEAD1 controls core metabolic processes and lipid accumulation in the liver.**

(A) Two-way Venn diagram intersects the number ($n$) of DEGs upon TEADap and TEADsf inhibitor treatments compared to control free fatty acid-fed (FFA + ) spheroids. (B) Line charts illustrate two clusters of $z$-score-scaled expression trends of unified DEGs ($n = 485$) in FFA+ spheroids without and with TEAD inhibitor treatments. The dashed black line indicates cluster centroid over all deregulated genes (gray). The number indicates genes per cluster (parentheses). (C) The schematic illustration displays core pathways altered by TEAD inhibition in FFA+ spheroids. Circles show individual genes in TEADap-treated compared to the untreated condition (blue: reduced, red: increased $log_2$FC). Red arrow indicates TEADap-mediated activation of AMPK signaling. Sharp and blunt arrows show activation and inhibition, respectively. (D) Schematic illustration (top) and density plot (bottom) show average TEAD1-binding site distributions in promoters of 1000 random non-deregulated gene sets ($n = 435$). The average number of TEAD1-binding sites in promoters of deregulated genes upon TEAD1 inhibition (black line) and $P$ value are displayed (permutation tests). (E) The model illustrates systemic ER activation effects on liver lipid accumulation upon *TEAD1* gene expression. Increased lipid levels promote *TEAD1* gene locus remodeling with higher enhancer activity and *TEAD1* gene expression (left, yellow background). Systemic ER activation with estrogen treatments suppresses *TEAD1* gene locus activity. Consequently, reduced TEAD1 levels can rewire liver metabolism and decrease lipid accumulation. This effect can be recapitulated by impairing TEAD1 activity using small molecule inhibitors (right, red background).

six-week-old male and female C57BL/6J mice obtained from in-house breeding were fed a control (D12450J, 10% kcal fat, Research Diet) or high-fat diet (D12492, 60% kcal fat, Research Diet) ad libitum for 13 weeks ($n = 4$ per condition). Subsets of male mice on HFD were additionally injected intraperitoneally with the estrogenic ligands 17β-estradiol (E2, 0.5 mg/kg body weight, Sigma-Aldrich), 4,4',4"-(4-Propyl-[1*H*]-pyrazole-1,3,5-triyl)*tris*phenol (PPT, 2.5 mg/kg body weight, Tocris), 2,3-Bis(4-hydroxyphenyl) propionitrile (DPN, 5 mg/kg body weight, Tocris) and 4-(2-(3,5-dimethylisoxazol-4-yl)-1H-indol-3-yl)phenol (DIP, 10 mg/kg body weight) or given a sham injection every second day from week 10 to week 13 ($n = 4$ per condition). Ligand concentrations were chosen according to literature (González-Granillo et al, 2019; Kim et al, 2011; Frasor et al, 2003). The ligands were diluted in 55% water, 40% PEG400 and 5% DMSO. Mice in each group were descended from different parents and were housed in at least two different cages at 20 °C and sacrificed at Zeitgeber time three to four. Upon sacrifice, blood glucose was measured after 2 h fasting with a glucometer (Accu-Chek) and livers of C57Bl/6J mice were dissected and washed with phosphate-buffered saline (PBS). Livers were either cross-linked for ChIP-seq, embedded for histology or flash-frozen in liquid nitrogen for RNA-seq.

## Liver histology of murine liver sections

Formalin-fixed and paraffin-embedded livers were processed into 3 μm thick sections, before staining with hematoxylin & eosin (Mayers Hematoxylin Plus #01825 and Eosin ready-made 0.2% solution #01650) according to standard histological procedures for the assessment of the liver histology.

## Cell culture

HepG2 and AML12 cell lines were obtained from the American Type Culture Collection with certified genotype and were regularly tested for mycoplasma (Eurofins Genomics). HepG2 cells were cultured in Dulbecco's Modified Eagle Medium (DMEM) supplemented with 10% fetal bovine serum (FBS, Hyclone, GE healthcare) and 1% penicillin–streptomycin (PS, Sigma-Aldrich) while AML12 cells in DMEM/F-12 (Gibco) supplemented with 10% FBS, 1% PS, 1% Insulin-Transferrin-Selenium Sodium Pyruvate (Gibco) and 40 ng/mL Dexamethasone (Sigma-Aldrich) in T75 flasks at 37 °C and 5% $CO_2$ atmosphere. Cells were passaged at a 1:6 ratio twice (HepG2) and three (AML12) times a week by aspirating the medium, gently washing the cells with PBS without $Mg^{2+}$ (Sigma-

Aldrich) and then detached using 2 mL of trypsin-EDTA solution (Sigma-Aldrich) for 3–5 min. Trypsin was inactivated with 8–10 mL of culture medium before passaging to a new flask.

## Primary human hepatocyte spheroid culturing

Spheroids were formed by seeding cryopreserved primary human hepatocytes (PHH) of a male donor in ultra-low attachment 96-well plates (Corning) as previously described (Bell et al, 2016). For spheroid treatments, free fatty acids were conjugated to 10% bovine serum albumin at a molar 1:5 ratio for 2 h at 40 °C. Formed spheroids were treated with 240 μM oleic acid and 240 μM palmitic acid along with 100 nM of either TEAD autopalmitoylation (TEADap, VT-104) (Tang et al, 2021) or TEAD surface inhibitor (TEADsf) (Bordas et al, 2021) inhibitors for 5 days. Intracellular lipid content was assessed using the AdipoRed Assay Reagent (Lonza).

## siRNA-mediated TEAD1/Tead1 knockdown

Confluent HepG2 (60–70%) or AML12 (80–90%) cells were trypsinized and electroporated with siRNAs targeting either TEAD1/Tead1 (SMARTpool, ON-TARGETplus™, Horizon Discovery) or a control nontargeting siRNA pool (ON-TARGETplus™, Horizon Discovery). After washing cells once with OptiMEM (Invitrogen), 2 million cells were resuspended in 200 μl OptiMEM and incubated for 3 min with 2 μg (7.5 μL of a 20 μM stock) siRNA in a 4 mm cuvette (Bio-Rad) before being pulsed at 300 V, 250 μF, in a Genepulser II (Bio-Rad). Immediately after electroporation, the cells were transferred to pre-heated (37 °C) phenol red-free DMEM (HepG2) or DMEM/F-12 (AML12) culture medium without antibiotics. Cells were collected at day 4 to determine knockdown efficiency and microscopy, and at day 5 for Seahorse analysis.

## Microscopic LD quantification

Two days after electroporation, 30,000 AML12 cells transfected with siNT or si*Tead1* were seeded into ibiTreat eight-well coverslips (ibidi). The following day, LDs were stained with LipidTOX Red (Thermo Fisher Scientific) 1:6250 (v:v) and nuclei with NucBlue (Thermo Fisher Scientific) 1:62.5 (v:v). After incubation at 37 °C and 5% $CO_2$ for 20 min, the cells were washed twice with Leibovitz's L15 medium. Images were acquired using a LSM780 confocal microscope (Zeiss) with a Zeiss C-APOCHROMAT water

immersion objective lens (40×/1.2). Imaging was performed at 37 °C and the sample IDs concealed (blinding). NucBlue and LipidTOX Red were excited using 405 nm and 640 nm laser lines, respectively. Image analysis was carried out using ImageJ. Nuclei were identified and subsequently counted by masking the NucBlue channel after applying a 3-pixel mean filter. Individual LDs were located by identifying local intensity maxima after applying a 2-pixel mean filter in the LD channel. To quantify the average number of LDs per cell, the total number of identified LDs in each image was divided by the nucleus number.

## Seahorse assay

Metabolic flux analysis was carried out on HepG2 cells using Seahorse XF96 Extracellular Flux Analyzer (Agilent). 15,000 cells were seeded the day before the experiment, and medium was changed to XF DMEM-based medium containing 2 mM GlutaMax, 25 mM glucose and 1 mM pyruvate on the day of the experiment and incubated at 37 °C without $CO_2$ for 1 h prior to the experiment. The oxygen consumption rate was measured at and following injection of oligomycin (1 μM final), FCCP (0.5–1.5 μM), and mixture of rotenone and antimycin A (4 μM). Data were normalized on the number of cells per well and against basal oxygen consumption rate. Cell number was normalized by nuclear staining (Hoechst, Molecular probes) for 10 min followed by imaging each well using BD pathway 855 (BD Biosciences) with a 10× objective and montage 5 × 4. Cell number was counted with Cell profiler software.

## RNA isolation and DNase treatment

Approximately 20 mg of flash-frozen liver tissue was homogenized in 700 μL QIAzol (QIAGEN) using a TissueLyzer II (QIAGEN, 2 min, 25 Hz, two times). The samples were incubated at room temperature for 5 min, before adding 140 μL chloroform (Sigma-Aldrich). This mixture was shaken for 15 s, incubated for 3 min, and centrifuged at 9000 × *g* at 4 °C for 5 min. The aqueous phase was carefully transferred to a new tube and an equal volume of isopropanol was added. This mixture was incubated at room temperature for 10 min, before centrifugation at 20,000 × *g* and 4 °C for 10 min. The supernatant was removed and the pellet washed twice with 70% ethanol, air-dried, and resuspended in water. The isolated RNA was DNase-treated using the Turbo DNase Kit (Thermo Fisher Scientific) according to the manufacturer's instructions. In brief, 10 μg of RNA was treated with 2U DNase and 40U RNaseOUT (Thermo Fisher Scientific) at 37 °C for 30 min. DNase-treated RNA from mouse livers was incubated with DNase inactivation reagent (Turbo DNase kit) for 5 min under constant homogenization. The sample was centrifuged at 10,000 × *g* for 2 min to remove the inactivation reagent. To purify the obtained DNase-treated RNA, the RNA was diluted to 130 μL with water, before adding 20 μL sodium acetate (Thermo Fisher Scientific, 3 M, pH 5.2), 1 μL GlycoBlue (Thermo Fisher Scientific) and 600 μL ice-cold 99.8% ethanol. Next, RNA was precipitated at −80 °C overnight, before centrifugation at 20,000 × *g* for 30 min, washing the pellet twice with 70% ethanol, air-drying and resuspending in water. DNase-treated RNA from liver spheroids was purified using an RNA clean and concentrator kit according to the manufacturer's instructions (Zymo research). The RNA quality was assessed on a Bioanalyzer 2100 device using RNA Nano chips (Agilent Technologies) and only high-quality RNAs (RIN > 6.5) were used for RNA-seq.

## RNA sequencing and data processing

Strand-specific RNA libraries ($n = 4$ mice per condition) were generated using the NEBNext Ultra II stranded library kit (New England Biolabs) combined with polyA-coupled beads (New England Biolabs) according to the manufacturer's instructions. The library quality was assessed on a Bioanalyzer 2100 device using DNA High Sensitivity chips (Agilent Technologies) and quantified using a KAPA library quantification kit (Roche). cDNA libraries were subsequently sequenced on an Illumina NextSeq 500 device using a paired-end high-output kit (75 + 75 cycles for mouse, 40 + 40 cycles for PHH spheroids). Reads were trimmed (Trimmomatic v0.36 for mouse, fastp v0.23.2 for PHH spheroids) and filtered for non-ribosomal RNA by mapping to a custom rRNA reference (HISAT2 v2.1 for mouse (Kim et al, 2019), SortMeRNA v4.3.6 for PHH spheroids). Non-aligned reads were further mapped to the mm10 mouse reference genome retrieved from GENCODE vM23 (HISAT2, GRCm38.p6) or Ensembl release 109 in the case of PHH spheroids (HISAT2 v2.2.1). Generated SAM files were converted to BAM files and consequently processed (SAMtools v1.9 for mouse, SAMtools v1.16.1 for PHH spheroids) (Li et al, 2009). bedGraph files were generated using HOMER (v4.10) (Heinz et al, 2010) for mouse or bedtools (v2.30.0) for PHH spheroids. Count tables were generated using SubRead (v1.5.2 for mouse, v2.0.3 for PHH spheroids) (Liao et al, 2013).

## Differential gene expression analysis

Differential gene expression analysis for mouse RNA-seq data was performed using DESeq2 (v1.30.0, default model) (Love et al, 2014) and edgeR (v3.32.1, glmFit model) (Robinson et al, 2010). Genes which were found to be differentially expressed in both analyses were considered for further analysis. Human spheroid RNA-seq data was analyzed with DESeq2 (v1.38.3, default model).

## Transcriptomic signal-to-noise ratio

Transcriptome-wide differences across conditions were measured unbiased by using a transcriptome-based signal-to-noise ratio (tSNR) as described previously (Lopes-Ramos et al, 2020). For this, the Euclidean metric was used as a measure of distance across transcriptomes. The signal was defined as the distance between the averaged transcriptomes of two groups while the noise was defined based on the total within-group variation observed (i.e., the dispersion of distance measurements of each sample transcriptome to the group average), expressed as:

$$tSNR(X, Y) = \frac{\overline{X} - \overline{Y}_2}{\sqrt{\frac{\sigma_X^2}{N} + \frac{\sigma_Y^2}{M}}}$$

Here, $\overline{X}$ and $\overline{Y}$ are the averaged transcriptomes, $N$ and $M$ indicate the sample number, and $\sigma_X^2$ and $\sigma_Y^2$ represent the intragroup variance for $X$ and $Y$ groups, respectively.

## Gene clustering and overrepresentation analysis

To identify shared gene expression patterns across the different diet and agonist-treated conditions, we applied a soft clustering strategy using the Mfuzz R package (v2.48.0) (Kumar & Futschik, 2007). Normalized gene expression values were z-score scaled ($\mu$=0; sd=1) and the optimal number of clusters was determined after testing for different numbers of clusters. Gene ontology (GO) and Kyoto Encyclopedia of Genes and Genomes (KEGG, release 106) analyses on selected gene sets were performed using hypergeometric tests with the hypeR (Federico and Monti, 2020) or clusterProfiler (Yu et al, 2012) R packages. GO biological process annotations were retrieved from the MGI database (v6.16; 03-2021), org.Mm.eg.db (v3.12.0 and v3.16.0) or org.Hs.eg.db (v3.16.0), and enriched terms were established using a $q$ value threshold of 0.05 and a custom gene background. The rrvgo (Sayols, 2023) package was used to cluster and produce visual representations of the overrepresented terms. Semantic similarities between terms were calculated using the Wang method and a similarity threshold of 0.9 was used for determining GO clusters.

## Pathway enrichment and network analysis

Gene set enrichment analysis (GSEA) was used to identify enriched Reactome pathways between conditions. The fgsea R package (v1.14.0) was run with gene lists ranked according to the signed $\log_{10} p$ values obtained from DESeq2 and pathway sizes were limited to a range of 10-500 genes relative to the background. Here, a $q$ value threshold of 0.05 was applied for enriched pathway selection. Network visualizations of all enriched pathways were generated using Cytoscape (v3.8.2) (Shannon et al, 2003). Individual pathways were connected according to their similarity ($s > 0.5$) and clusters of the interconnected pathways were produced using the GLay community clustering algorithm from clusterMaker (Morris et al, 2011). To uncover shared or divergent trends across different processes, pathway clusters were correlated based on their average normalized enrichment score (NES) and connections were filtered to those with $|r| > 0.9$. The similarity score between pathways used is a metric of both the jaccard similarity and overlap coefficients, calculated as:

$$s(A, B) = \frac{\frac{|A \cap B|}{|A \cup B|} + \frac{|A \cap B|}{\min(|A|, |B|)}}{2}$$

Where $A$ and $B$ represent the two sets of genes that are part of the pathways being compared.

## Single-cell data analysis

Preprocessed public single-cell and spatial transcriptomics datasets and annotations were retrieved from the Liver Cell Atlas (Guilliams et al, 2022). Given our gene signatures were defined in male mice, only cells originating from male mice samples were used in the analysis and primary cells were removed. Accordingly, only cells obtained from male macaque and human were considered. Cell type composition analyses were conducted in R using Seurat (v4.0.2) (Satija et al, 2015). Enrichment scores for the relevant ER

activation signature gene sets and Reactome pathway clusters identified were calculated using pagoda2 (v1.0.2). Up to 5000 cells for each annotated cell type were subsampled for the analysis. Pathway activity scores were aggregated at the cell type level by averaging the enrichment values of all individual cells annotated for a given cell type cluster and condition. To make pathway activity scores comparable, the scores were scaled to a 0–1 range using the min-max scaling method across all cell types for each pathway. Changes in pathway activity were measured as the difference between control and HFD scores for each cell type.

## ChIP-sequencing and data analysis

Formaldehyde-fixed livers ($n = 3$ mice per condition) were homogenized using a douncer and washed twice with ice-cold PBS. Nuclei were prepared as previously described (Schmidt et al, 2009) and sonicated using a Sonics Vibra cell VCX 750 set to 32% duty cycle for 30 cycles (30 s on, 59 s off). ChIP was performed using antibodies against H3K27ac (Abcam #4729, 5 µg) and H3K4me3 (monoclonal, Merck 05-1339, 5 µg) as previously described (Schmidt et al, 2009). Of note, ChIP-seq experiments were performed in two batches using H3K27ac antibodies from different lots, which could introduce batch-driven variation. Libraries from immunoprecipitated DNA were generated using the SMARTer ThruPLEX DNA-seq Kit (Takara Bio), size-selected and quality assessed by Bioanalyzer DNA High Sensitivity chips (Agilent Technologies) according to manufacturer's protocols. Libraries were quantified using KAPA quantification kit (Roche) and sequenced on an Illumina NextSeq 500 device using a single-end (75 cycles) high-output kit. Reads were mapped to the mouse reference genome (GRCm38.p6/mm10) using bowtie2 (v2.3.5.1) (Langmead and Salzberg, 2012), processed and sorted (SAMtools v1.12), regions masked (NGSUtils v0.5.9) and duplicate reads were removed and indexed (SAMtools v1.12). Peaks were identified using MACS (v2.2.6) (Zhang et al, 2008). bedGraph files were generated using deepTools (v3.3.2) (Ramírez et al, 2014) and differentially bound peaks were determined using DiffBind (v3.0.15) with the threshold FDR < 0.05 and $|\log_2 FC| > 0.585$. Selected regions were annotated using ChIPpeakAnno (v3.24.2) (Zhu et al, 2010). Raw H3K4me1 (E-MTAB-7127), CTCF (E-MTAB-437) and ERα (GSE49993) ChIP-seq data were retrieved and processed.

## Quantification of H3K27ac signals in differentially acetylated regions

BED file containing differentially acetylated promoters and enhancers ($\pm 200$ bp from the peak center) was converted into SAF format. H3K27ac BAM files and SAF annotation file were used to generate a count table normalized by counts per million (SubRead v2.0.0).

## Enhancer-gene pair analysis

The closest transcription start sites (one upstream, two downstream) to each differentially acetylated enhancer were determined using BEDOPS (Neph et al, 2012) closest-features (v2.4.39). H3K27ac and gene expression (TPM) of single replicates were correlated using Pearson correlation. Only enhancer-gene pairs

with H3K27ac to gene expression correlation of $P < 0.01$ were considered. Enhancer-gene pairs containing genes recovered by estrogenic ligand treatments were further analyzed. CTCF motif orientation (MA0139.1) in the mouse genome (mm10) was determined using FIMO (MEME Suite) (Grant et al, 2011). To identify potential canonical or non-canonical CTCF-mediated chromatin loops, we filtered for enhancers harboring an upstream CTCF peak (plus-strand oriented motif for canonical and minus-strand oriented motif for non-canonical loops, within 50 kb) when the paired genes were located downstream, and for downstream CTCF peaks (minus-strand oriented motif for canonical or plus-strand oriented motif for non-canonical loops, 50 kb) when the paired genes were located upstream of the enhancer. Promoter-capture Hi-C data (GSE155153, Zeitgeber time 0) were lifted over to mm10 to generate contact maps (UCSC LiftOver).

## MASLD patient comparison

Human orthologs of the genes part of estrogen-sensitive enhancer-gene pairs in mice ($n = 49$) were determined using Ensembl release 105. The MASLD cohort data (Govaere et al, 2020) (gender-balanced) was retrieved from Gene Expression Omnibus (GSE135251) and normalized to counts per million (CPM), scaled and centered ($z$-score). Only human orthologs with gene expression CPM > 0.5 were considered ($n = 45$). For NAS and fibrosis categorization, classes were defined as early (NAS0-1, F0-1), mid (NAS2-6, F2), and advanced (NAS7-8, F3-4).

## Transcription factor motif search

Genome-wide transcription factor binding sites (TFBS) for TEAD1 were identified using PWMscan (Ambrosini et al, 2018). The mononucleotide position weight matrix for the TEAD1-binding motif was retrieved from the HOCOMOCO v11 database (Kulakovskiy et al, 2013) for establishing TFBS a background nucleotide composition ($A = 0.29$, $C = 0.21$, $G = 0.21$, $T = 0.29$) and $P$ value cutoff of 0.00001 were set. Only TFBS in gene promoters were considered, defined as those within 1.5 kb upstream and 0.5 kb downstream of transcription start sites. The promoter region with the highest number of TFBS was assigned to each gene.

## Use of standardized official symbols

We use HUGO (Human Genome Organization) Gene Nomenclature Committee-approved official symbols (or root symbols) for genes and gene products, all of which are described at www.genenames.org. Gene symbols are italicized, whereas symbols for gene products are not italicized.

## Statistics

All analyses were conducted in R (4.0 or 4.2). The Shapiro–Wilk test was used to assess the normality of the data.

## Exclusion criteria

After close inspection of the transcriptomic signatures derived from our mouse experiments, one male mouse fed a HFD and injected with PPT was excluded from differential gene expression and downstream analyses given its extreme outlier status indicative of a failed treatment intervention.

## Data availability

Microscopic images and imageJ macro are available under: https://figshare.com/s/02d28cfc9a8ebb1f39c7. Raw and processed sequencing data generated in this study have been submitted to ArrayExpress. ChIP-seq data (H3K4me3/H3K27ac) in mouse: ArrayExpress E-MTAB-11929. Bulk RNA-seq data in mouse: ArrayExpress E-MTAB-11833. Bulk RNA-seq data of human PHH: ArrayExpress E-MTAB-13207. Detailed descriptions of bioinformatics analyses and scripts to reproduce the results are available in the Supplemental materials and methods and on GitHub: https://github.com/carlga/MAFLD_ER_agonists. Publicly available data were retrieved from the following domains: H3K4me1 ChIP-seq: ArrayExpress, E-MTAB-7127. CTCF ChIP-seq: ArrayExpress, E-MTAB-437. TEAD1, YAP and TAZ ChIP-seq: Gene Expression Omnibus, GSE163458. Mouse promoter captures Hi-C (CHi-C): Gene Expression Omnibus, GSE155153. ERα ChIP-seq: Gene Expression Omnibus, GSE49993. MASLD patient cohort RNA-seq: Gene Expression Omnibus, GSE135251. Liver single-cell and spatial transcriptomics data: www.livercellatlas.org.

## Peer review information

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

## Acknowledgements

We would like to thank the group members of the laboratories of Claudia Kutter, Marc Friedländer, and Vicent Pelechano for helpful feedback regarding the experimental procedures, data analysis, and data presentation, as well as Noah Moruzzi for assistance with the Seahorse experiment. We thank the Liver Cell Atlas, EPoS, and LITMUS consortium for sharing invaluable data with the research community. This work was supported by the Knut & Alice Wallenberg foundation (KAW 2016.0174, CK), Ruth & Richard Julin foundation (2017-00358, 2018-00328, 2020-00294, 2022-00283, 2023-00162, CK; 2021-00158, VML); SFO-SciLifeLab fellowship (SFO_004, CK), Swedish Research Council (2019-05165, 2023-02780, CK; 2019-01837, 2021-02801, VML; 2022-00901, CW), KI-KID (2018-00904, 2021-00308, CK; 2-3591/2014, 2018-00947, CW), KI-KIRI (2022-02535, ES, CK), KI-SRP Diabetes (2023, CK), Lillian Sagen & Curt Ericsson research foundation (2021-00427, CK), Gösta Milton's research foundation (2021-00527, CK), Robert Lundberg's Memorial Foundation (2022-01158, CS; 2022-01159, KG), Chinese Scholarship Council (201700260271; KG, CK), ERASMUS+ (20180716, CS), Novo Nordisk Foundation (NNF14OC0010705, MKA), Lisa and Johan Grönbergs Foundation (2019-00173, MKA), Astra Zeneca (ICMC, MKA), the National Academic Infrastructure for Supercomputing in Sweden (NAISS) at UPPMAX (storage: 2020/15-225, 2021/23-691, 2023/23-15; compute: 2020/16-291, 2021/22-960, 2023/22-36) and National Microscopy Infrastructure, NMI (VR-RFI 2016-00968, CK).

## Author contributions

**Christian Sommerauer**: Data curation; Software; Formal analysis; Validation; Investigation; Visualization; Methodology; Writing—original draft; Writing—review and editing. **Carlos J Gallardo-Dodd**: Data curation; Software; Formal analysis; Validation; Investigation; Visualization; Methodology; Writing—original draft; Writing—review and editing. **Christina Savva**: Investigation; Writing—review and editing. **Linnea Hases**: Investigation; Writing—review and editing. **Madeleine Birgersson**: Investigation; Writing—review and editing. **Rajitha Indukuri**: Investigation; Writing—review and editing. **Joanne X Shen**: Investigation; Writing—review and editing. **Pablo Carravilla**: Investigation; Writing—review and editing. **Keyi Geng**: Investigation; Writing—review and editing. **Jonas Nørskov Søndergaard**: Investigation; Writing—review and editing. **Clàudia Ferrer-Aumatell**: Investigation; Writing—review and editing. **Grégoire Mercier**: Investigation; Writing—review and editing. **Erdinc Sezgin**: Conceptualization; Funding acquisition; Project administration; Writing—review and editing. **Marion Korach-André**: Conceptualization; Supervision; Funding acquisition; Project administration; Writing—review and editing. **Carl Petersson**: Resources; Writing—review and editing. **Hannes Hagström**: Conceptualization; Resources; Supervision; Funding acquisition; Project administration; Writing—review and editing. **Volker M Lauschke**: Conceptualization; Resources; Funding acquisition; Project administration; Writing—review and editing. **Amena Archer**: Conceptualization; Resources; Investigation; Project administration; Writing—review and editing. **Cecilia Williams**: Conceptualization; Resources; Supervision; Funding acquisition; Project administration; Writing—review and editing. **Claudia Kutter**: Conceptualization; Resources; Data curation; Formal analysis; Supervision; Funding acquisition; Investigation; Writing—original draft; Project administration; Writing—review and editing.

## Funding

## Disclosure and competing interests statement

CP is an employee of the Healthcare Business of Merck KGaA (Darmstadt, Germany). HH's institutions have received research funding from Astra

Zeneca, EchoSens, Gilead, Intercept, MSD, and Pfizer, all outside this study. HH has served as a consultant for Astra Zeneca and has been part of hepatic events adjudication committees for KOWA and GW Pharma. VML is a co-founder, CEO, and shareholder of HepaPredict AB. The remaining authors declare no competing interests.

# Expanded View Figures

**Figure EV1.  Male but not female mice exhibit HFD-induced liver steatosis.**

(**A–C**) Bar charts display physiological parameters assessed upon sacrifice in female (f) and male (m) mice (n = 4 mice per condition, +SD), which were previously examined (Hases et al, 2020). (**A**) Total weight, (**B**) liver weight and (**C**) blood glucose after 2 h fasting were measured. Color gradient indicates female and male mice on different diets (CD or HFD) and male mice upon ERα/β-agonist treatments (DPN, DIP, E2 and PPT). *P* values highlight non-significant, and asterisks indicate significant differences (*$P < 0.05$, **<0.01, ***$P < 0.005$, one-way ANOVA followed by Tukey's post-hoc test). (**D**) Hepatic cross-sections of four female and male mice on different diets (CD or HFD) and HFD males treated with an ER-agonist (DPN, DIP, E2 or PPT) were stained with hematoxylin and eosin. Images shown in Fig. 1B are highlighted in black boxes. Scale bar: 100 μm. (**E**) Two-way Venn diagram (left) shows the intersection and number (*n*) of genes deregulated exclusively in HFD male (top) or HFD female (bottom) mice compared to CD or deregulated in both sexes (middle). Bubble plots (right) show the top eight gene ontology terms (biological processes) for each intersected gene set. One-sided Fisher's exact test with Benjamini–Hochberg correction was used for the overrepresentation analysis.

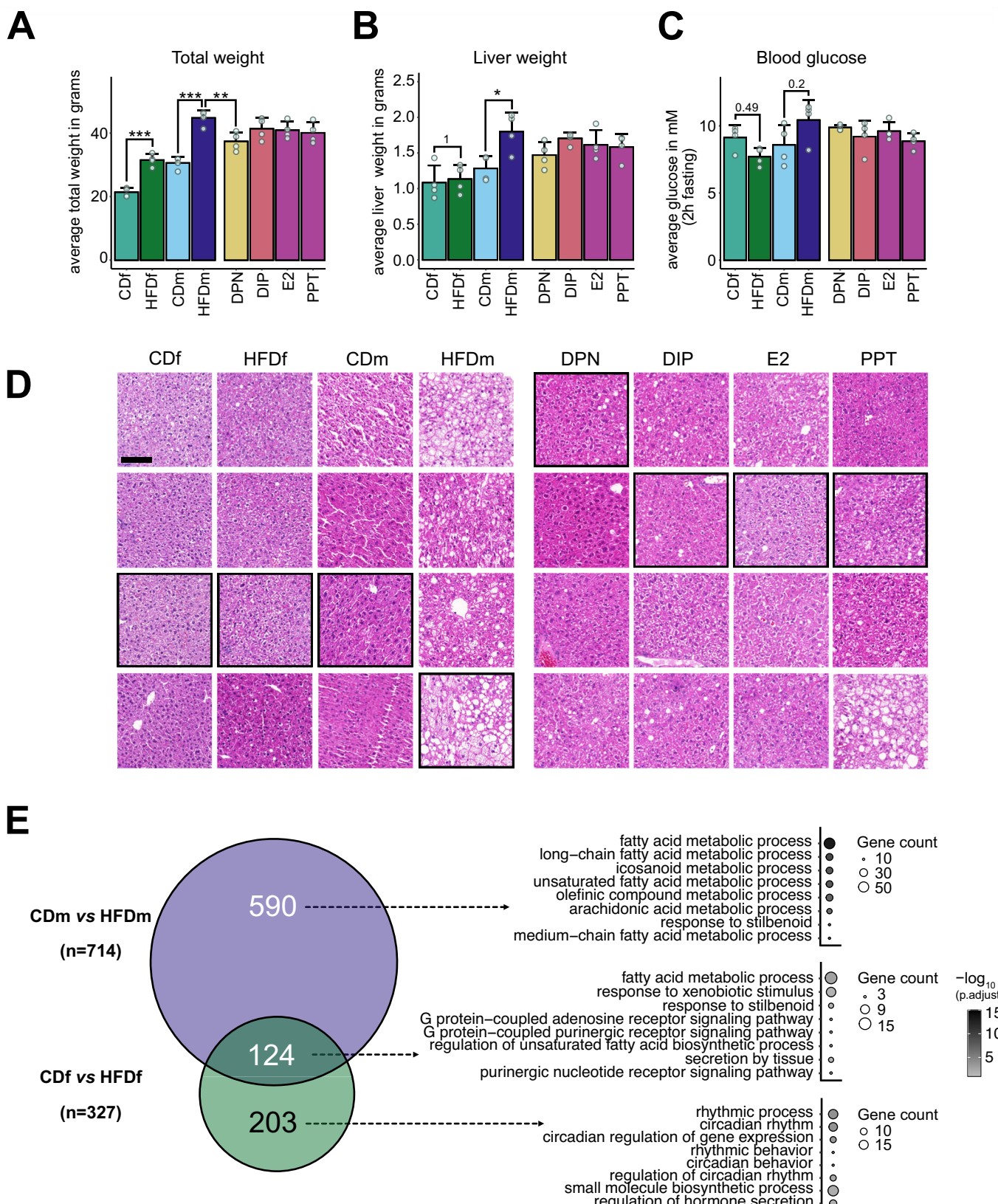

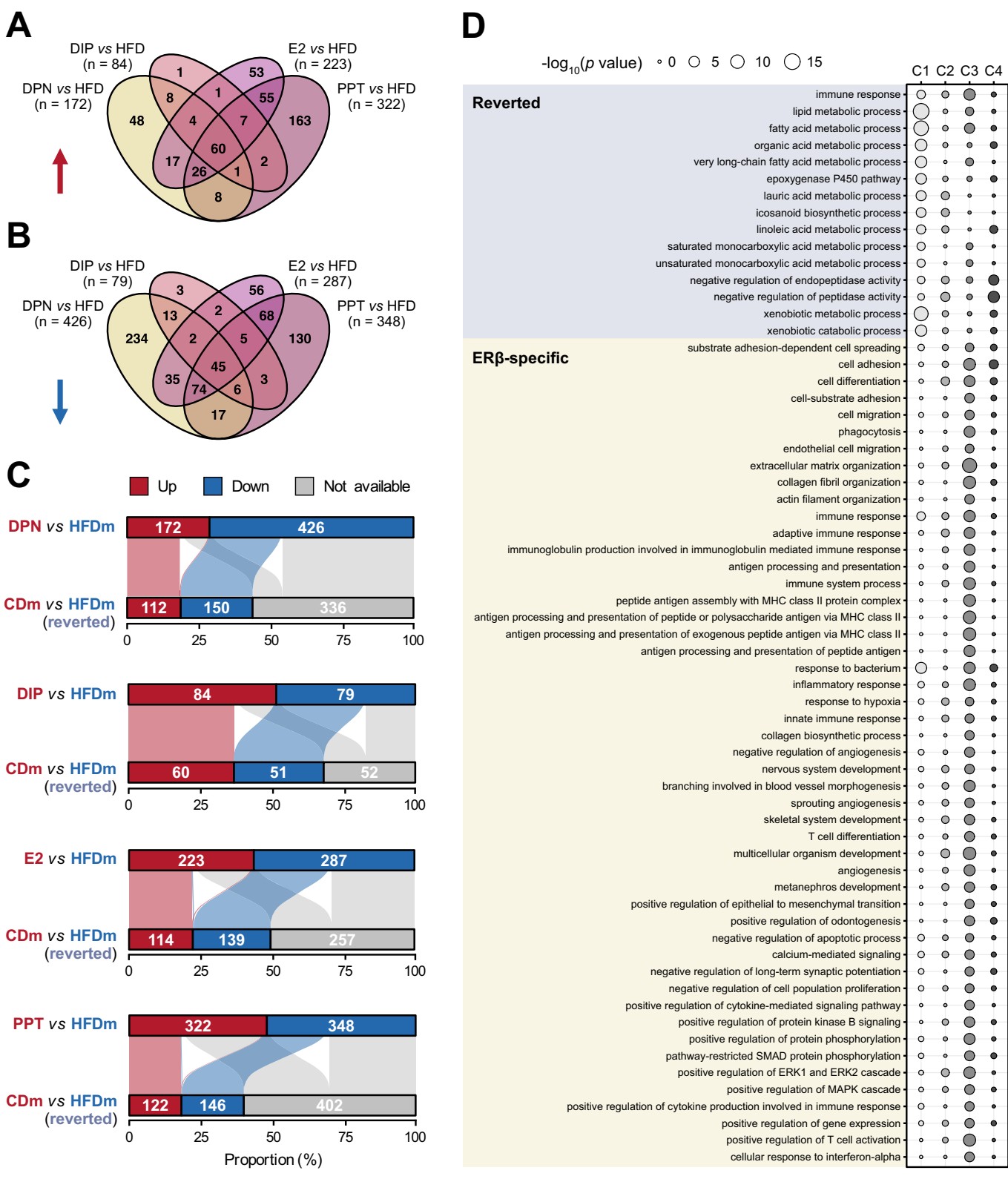

◀ **Figure EV2.** **ERα/β-agonist treatments control molecular responses.**

(A, B) Four-way Venn diagrams show intersections and numbers (*n*) of gene sets that are either (A) upregulated (red arrow) or (B) downregulated (blue arrow) by ER-agonist treatments compared to HFD in male mice (*n* = 4 mice per condition). (C) Horizontal bar plots (top) display the proportional frequency of up- (red) and downregulated (blue) genes in male mice on HFD compared to HFD male mice treated with an ERα/β-agonist (DPN, DIP, E2 or PPT). Horizontal bar plots (bottom) show the respective occurrences of the deregulated genes upon ER-agonist treatment in a diet comparison of male CD *versus* HFD. Genes significantly higher in CD male (red), higher in HFD male (blue) or unaltered (gray) are shown. Alluvial line width connecting upper and lower side indicate how many HFD-deregulated genes are recovered upon treatment (from red to red or blue to blue). Genes indicated in light gray are unchanged by HFD. Genes changed in both comparisons but not recovered upon treatment (from red to blue or blue to red) were categorized as non-reverted. (D) Bubble plot displays significantly enriched gene ontology terms (biological processes) per cluster (Fig. 2A, B) in reverted (denim) and ERβ-specific (ocher) gene sets. Circle color indicates the corresponding cluster, and circle size represent statistical significance of the gene ontology term (-$\log_{10}$ *P* value, low: narrow, high: wide). Hypergeometric test with Benjamini–Hochberg correction was used for the overrepresentation analysis.

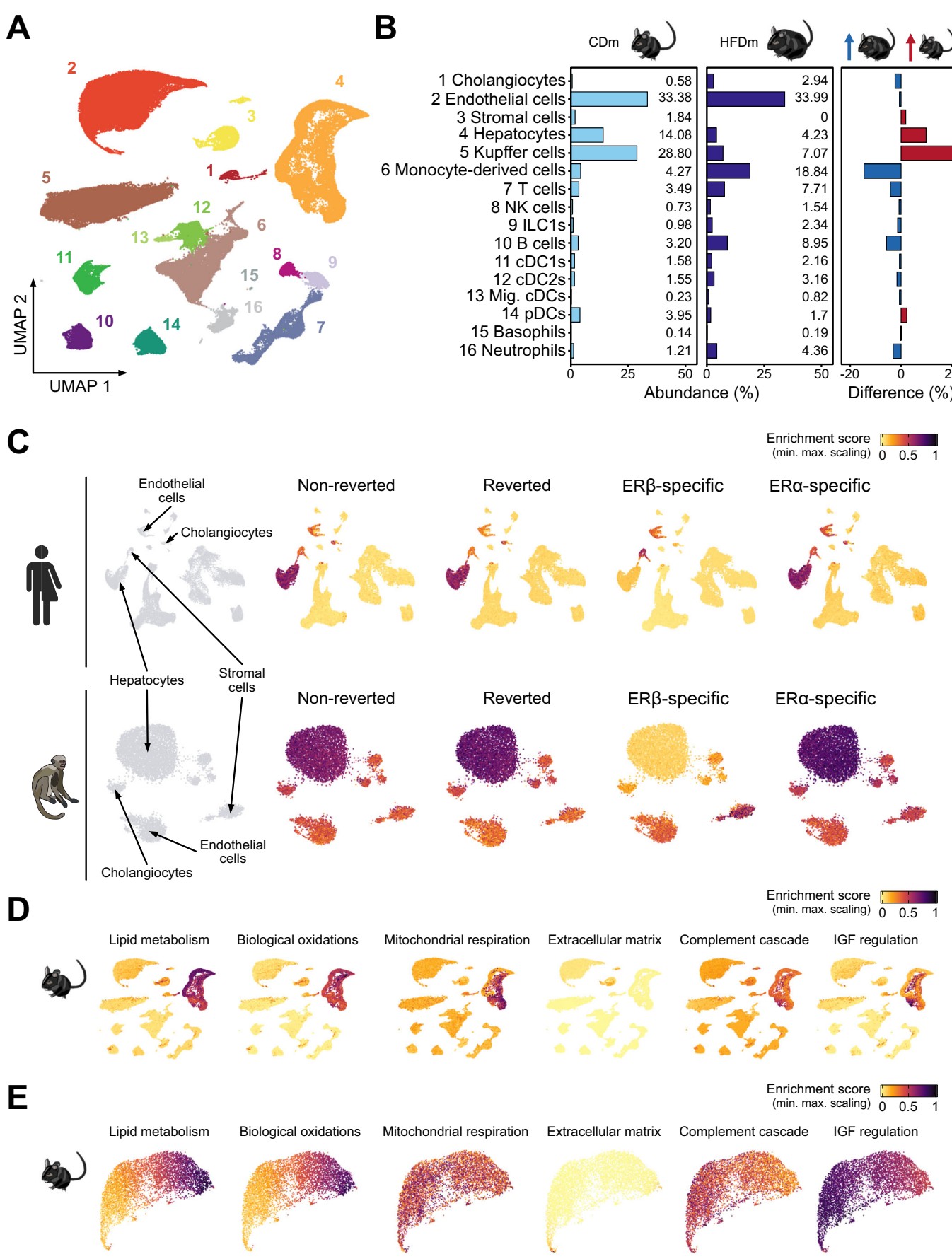

**Figure EV3.  ER-responsive molecular gene signatures show cell type-specificity and are maintained in primates.**

(A) Single-cell map projected in UMAP space displays reference annotation of liver cell types (numbered according to Fig. EV3B). (B) Bar plots show the relative abundance in percent of distinct liver cell populations in control (left) and MASLD (middle) mice as well as their difference (right). Arrows indicate higher abundance in control (red) and MASLD (blue) mice. (C) Enrichment maps present liver cell type specificities of murine gene sets (defined in Fig. 2B) in human (top) and macaque (bottom) livers. Four relevant mouse- and primate-maintained cell types are labeled. (D) Enrichment maps show signal distribution of enriched pathways across liver cell types in mouse. (E) Spatial transcriptomics maps display liver zonation patters of the selected pathways. (C–E) Enrichment score: low: yellow, high: black.

                                                

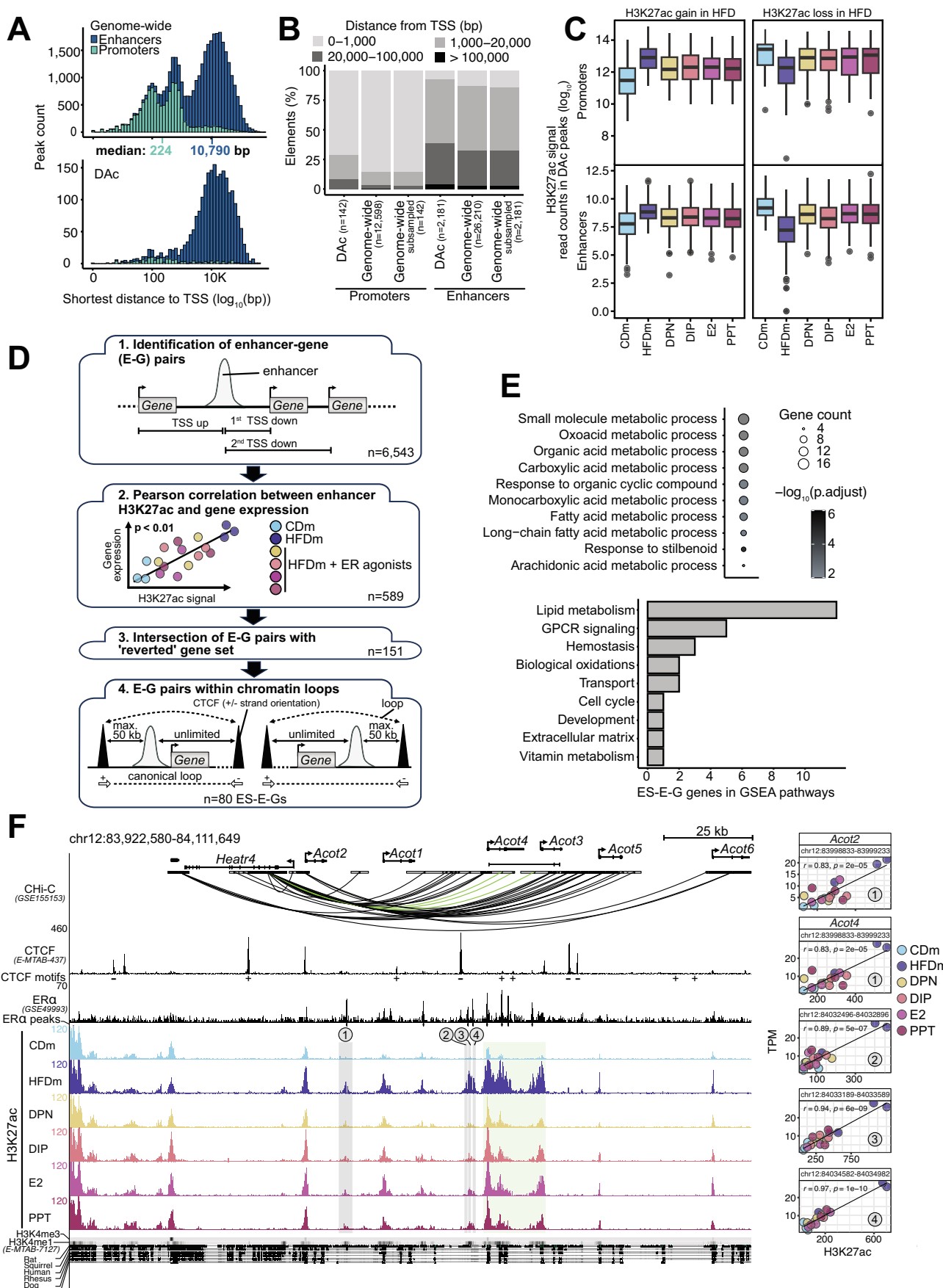

◀ **Figure EV4.   ERα/β-agonist treatments alter metabolic gene regulation.**

(A) Histograms show frequency of genome-wide (top) or differentially acetylated (DAc, bottom) promoters (teal) and enhancers (dark blue) (*y* axis) with respect of distance to the closest annotated transcription start site (TSS, $\log_{10}$ scale, *x* axis). Median distance from nearest annotated TSS to promoters (teal) and enhancers (dark blue) is highlighted. (B) Stacked bar plot indicates percentage of DAc or genome-wide (all or downscaled to the number of DAc) promoter and enhancer elements binned according to the distance to the closest annotated TSS in bp. The numbers (*n*) of DAc and all promoter and enhancer regions are specified in parenthesis. (C) Box plots illustrate number of normalized (1× genome coverage) reads in peaks ($\log_2$) for the same regions and diet comparisons as in Fig. 4A (*n* = 3 mice per condition). Each box represents interquartile range (IQR) divided by the median (horizontal line), and whiskers span a maximum of 1.5×IQR. Outliers (circles) are shown. (D) Workflow displays identification of estrogen-sensitive enhancer-gene pairs (ES-E-Gs). First, for each enhancer, which is DAc in CDm *versus* HFDm (*n* = 2181), the three closest TSS are determined. The directionality of nearby gene transcription is not considered, instead the distance to the TSS (*n* = 6543). Second, Pearson correlation (*r*) and significance between H3K27ac signal and gene expression is assessed for each E-G pair, within individual replicates, considering only correlations of *P* < 0.01. Colors indicate male mice on different diets (CD or HFD) and upon ERα/β-agonist treatments. Third, E-G pairs with genes identified as ER-agonist treatment-reverted at the transcript level (Fig. 2B) and fourth, those with a CTCF peak within 50 kb under consideration of strand orientation, are selected. Numbers (*n*) of E-G pairs are displayed. (E) Bubble plot (top) shows the top 12-enriched gene ontology terms for the 49 genes part of ES-E-Gs (-$\log_{10}$ adjusted *p* value, low: narrow, high: wide). One-sided Fisher's exact test with Benjamini–Hochberg correction was used for the overrepresentation analysis. Horizontal bar plot (bottom) illustrates the contributions of the 49 genes to the GSEA pathway clusters (Fig. 3A). (F) Genome browser view (left) shows genomic regions (mm10) around *Acot2* and *Acot4* gene loci. Black boxes represent exons and UTRs. Arrows indicate directionality of gene transcription. Scale bar shows length of genomic regions in kilobases (kb). Promoter-capture Hi-C (CHi-C) 3D connections are shown for the *Acot* gene loci as black and green arcs. Genomic regions are enriched for CTCF (black peaks) with CTCF motif orientations determined with FIMO (indicated by plus or minus symbols), ERα (black peaks) and significant ERα peaks (black insets), H3K27ac in livers of CDm, HFDm and ER-agonist-treated HFDm, H3K4me3 and H3K4me1 (horizonal gray bars, dark: high, light: low). One replicate per condition is shown. The *y* axis of each track specifies normalized read density in livers of male mice. Genomic location of enhancers (numbered from 1 to 4) paired with the *Acot* gene loci are highlighted (gray vertical boxes). An additional enhancer-rich region across the *Acot4* and *Acot3* gene loci is shown (green vertical box). The degree of genomic sequence conservation at base resolution across selected vertebrates is shown (conserved: black, not conserved: white). Scatter plots (right) correlate *Acot2* and *Acot4* gene expression (TPM, *y* axis) and their respective paired enhancers (H3K27ac signals, *x* axis) in livers of male mice on different diets and ER-agonist treatments. All replicates are shown. Enhancer coordinates (400 bp window around the enhancer summit), Pearson correlation coefficient (*r*) and significance (*p*) are indicated in each box.

                                          

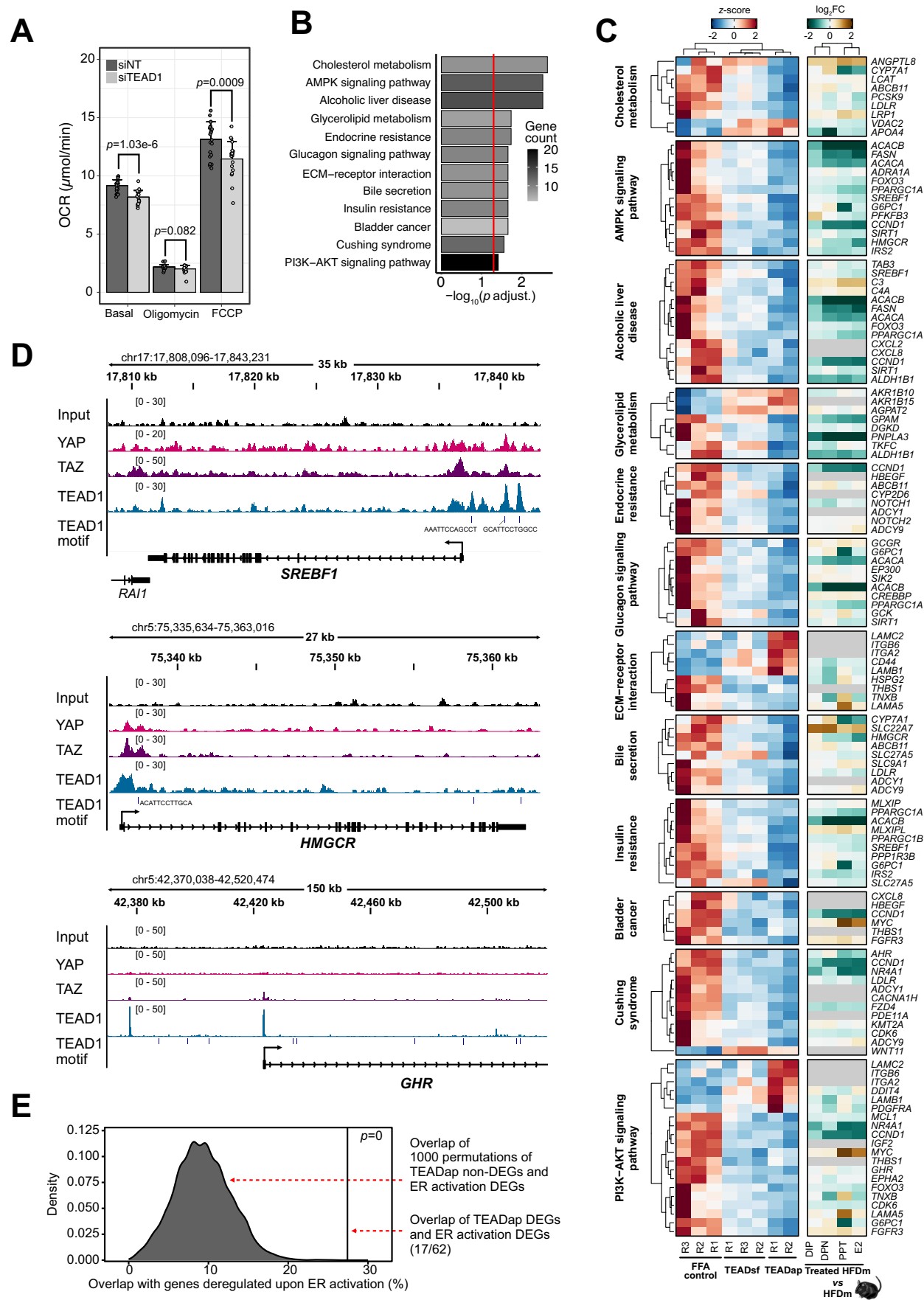

◀  **Figure EV5.   TEAD inhibition changes key cellular processes in the liver.**

(A) Bar plot displays mean of basal, minimum, and maximum oxygen consumption rate in HepG2 cells with control (siNT, dark gray) and siRNA-mediated *TEAD1*-KD (siTEAD1, light gray) ($n = 63$ per condition, +SD). Dots indicate single wells across two biological replicates. Significant *P* values are indicated (two-sided *t* test). (B) Bar plot shows the adjusted *P* values ($\log_{10}$) of the top 12 KEGG pathways enriched for genes deregulated upon TEADap treatment. Red vertical line indicates adjusted *P* value threshold ($P < 0.05$). Color gradient indicates gene number per pathway (low: gray, high: black). One-sided Fisher's exact test with Benjamini–Hochberg correction was used for the overrepresentation analysis. (C) Heatmap displays gene expression changes of the top 12 KEGG pathways altered upon TEADap treatment. Left panel indicates normalized gene expression (*z*-score) in free fatty acid-fed controls, TEADsf and TEADap treatments (low: blue, high: red). Right panel shows $\log_2$FC for ER-agonist treatments compared to untreated male mice on HFD (low: green, high: brown). Gene names follow human nomenclature. Ortholog absence in mouse is specified (gray). (D) Genome browser views (IGV, hg38) show genomic regions around *SREBF1, HMGCR* and *GHR* gene loci. Genomic locations and sizes are indicated. The *y* axis of each track specifies normalized ChIP-seq read density (parenthesis) of YAP (pink), TAZ (purple) or TEAD1 (blue) in HUVEC cells (GSE163458). Vertical lines indicate TEAD1-binding sites identified by motif search. Blue boxes represent exons and UTRs, connecting lines indicate intronic sequences. Arrows indicate directionality of gene transcription. (E) Density plot shows the distribution of overlaps between 1000 random sets of genes unchanged by TEADap treatment and genes changed by ER-agonist treatments falling within the top 12-enriched KEGG pathways (gray peak). The percentage of overlapping genes changed in both TEADap and ER-agonist treatments for the top 12-enriched KEGG pathways is displayed (black vertical line). *P* value is indicated (permutation tests).

                  