## [Peer Review File · Molecular Systems Biology]

Estrogen receptor activation remodels TEAD1 gene expression to alleviate hepatic steatosis

Christian Sommerauer, Carlos Gallardo-Dodd, Christina Savva, Linnea Hases, Madeleine Birgersson, Rajitha Indukuri, Joanne Shen, Pablo Carravilla, Keyi Geng, Jonas Nørskov Søndergaard, Clàudia Ferrer-Aumatell, Grégoire Mercier, Erdinc Sezgin, Marion Korach-André, Carl Petersson, Hannes Hagström, Volker Lauschke, Amena Archer, Cecilia Williams, and Claudia Kutter

Corresponding author(s): Claudia Kutter (claudia.kutter@ki.se)

Review Timeline:

Submission Date:	14th Sep 23
Editorial Decision:	11th Oct 23
Revision Received:	11th Jan 24
Editorial Decision:	30th Jan 24
Revision Received:	10th Feb 24
Accepted:	13th Feb 24

Editor: Maria Polychronidou

Transaction Report:

11th Oct 2023

Manuscript Number: MSB-2023-11999

Title: Estrogen receptor activation remodels TEAD1 gene expression to alleviate hepatic steatosis in NAFLD

Dear Claudia,

Thank you again for submitting your work to Molecular Systems Biology. We have now heard back from the three reviewers who agreed to evaluate your study. As you will see below the reviewers acknowledge that the presented findings seem relevant for the field. They do however raise a series of concerns which we would ask you to address in a revision.

I think that the issues raised by the reviewers are rather clear and I therefore see no need to repeat them here. All issues raised would need to be satisfactorily addressed. Please feel free to contact me in case you would like to discuss in further detail any of the issues raised. I would be happy to schedule a call.

On a more editorial level, we would ask you to address the following points:

- Please provide a .doc version of the manuscript text (including legends for the main figures and EV figures) and individual production quality figure files for the main Figures and EV Figures (one file per figure). Tables (if there are any) should be included in the manuscript text (at the very end), together with their description/legend.
 - We have replaced Supplementary Information by the Expanded View (EV format). In this case, all additional figures can be included in a PDF called Appendix. Appendix figures should be labeled and called out as: "Appendix Figure S1, Appendix Figure S2... Appendix Table S1..." etc. Each legend should be below the corresponding Figure/Table in the Appendix. Please include a Table of Contents in the beginning of the Appendix. For detailed instructions regarding expanded view please refer to our Author Guidelines: .
 - Supplemental Tables 1-14 should be provided as EV Tables (if shorter than one page) or EV Datasets (if complex/long). Please provide one file per EV Table/Dataset. Each file should include the description of the EV Table/Dataset in a separate tab.
 - Please provide a "standfirst text" summarizing the study in one or two sentences (approximately 250 characters), three to four "bullet points" highlighting the main findings and a "synopsis image" (550px width and max 400px height, jpeg format) to highlight the paper on our homepage.
 - All Materials and Methods need to be described in the main text. We would encourage you to use 'Structured Methods', our new Materials and Methods format. According to this format, the Materials and Methods section should include a Reagents and Tools Table (listing key reagents, experimental models, software and relevant equipment and including their sources and relevant identifiers) followed by a Methods and Protocols section in which we encourage the authors to describe their methods using a step-by-step protocol format with bullet points, to facilitate the adoption of the methodologies across labs. More information on how to adhere to this format as well as downloadable templates (.doc or .xls) for the Reagents and Tools Table can be found in our author guidelines: . An example of a Method paper with Structured Methods can be found here:
 - Please include a "Disclosure and Competing Interests Statement" in the main text.
 - Please include a Data availability section describing how the data, code etc. have been made available. This section needs to be formatted according to the example below:
The datasets and computer code produced in this study are available in the following databases:
 - Chip-Seq data: Gene Expression Omnibus GSE46748 (<https://www.ncbi.nlm.nih.gov/geo/query/acc.cgi?acc=GSE46748>)
 - Modeling computer scripts: GitHub (<https://github.com/SysBioChalmers/GECKO/releases/tag/v1.0>)
 - [data type]: [full name of the resource] [accession number/identifier] ([doi or URL or identifiers.org/DATABASE:ACCESSION])
 - For data quantification: please specify the name of the statistical test used to generate error bars and P values, the number (n) of independent experiments (specify technical or biological replicates) underlying each data point and the test used to calculate p-values in each figure legend. The figure legends should contain a basic description of n, P and the test applied. Graphs must include a description of the bars and the error bars (s.d., s.e.m.).
 - When you resubmit your manuscript, please download our CHECKLIST (<https://bit.ly/EMBOPressAuthorChecklist>) and include the completed form in your submission.
- *Please note* that the Author Checklist will be published alongside the paper as part of the transparent process (<https://www.embopress.org/page/journal/17444292/authorguide#transparentprocess>).

If you feel you can satisfactorily deal with these points and those listed by the referees, you may wish to submit a revised version

of your manuscript. Please attach a covering letter giving details of the way in which you have handled each of the points raised by the referees. A revised manuscript will be once again subject to review and you probably understand that we can give you no guarantee at this stage that the eventual outcome will be favorable.

Kind regards,

Maria

Maria Polychronidou, PhD
Senior Editor
Molecular Systems Biology

We realize that it is difficult to revise to a specific deadline. In the interest of protecting the conceptual advance provided by the work, we recommend a revision within 3 months (9th Jan 2024). Please discuss the revision progress ahead of this time with the editor if you require more time to complete the revisions. Use the link below to submit your revision:

IMPORTANT: When you send your revision, we will require the following items:

1. the manuscript text in LaTeX, RTF or MS Word format
2. a letter with a detailed description of the changes made in response to the referees. Please specify clearly the exact places in the text (pages and paragraphs) where each change has been made in response to each specific comment given
3. three to four 'bullet points' highlighting the main findings of your study
4. a short 'blurb' text summarizing in two sentences the study (max. 250 characters)
5. a 'thumbnail image' (550px width and max 400px height, Illustrator, PowerPoint or jpeg format), which can be used as 'visual title' for the synopsis section of your paper.

6. Please include an author contributions statement after the Acknowledgements section (see

<https://www.embopress.org/page/journal/17444292/authorguide>)

7. Please complete the CHECKLIST available at (<https://bit.ly/EMBOPressAuthorChecklist>).

Please note that the Author Checklist will be published alongside the paper as part of the transparent process

(<https://www.embopress.org/page/journal/17444292/authorguide#transparentprocess>).

See also figure legend guidelines: <https://www.embopress.org/page/journal/17444292/authorguide#figureformat>

9. Please note that corresponding authors are required to supply an ORCID ID for their name upon submission of a revised manuscript (EMBO Press signed a joint statement to encourage ORCID adoption).

(<https://www.embopress.org/page/journal/17444292/authorguide#editorialprocess>)

Currently, our records indicate that the ORCID for your account is 0000-0002-8047-0058.

Link Not Available

The system will prompt you to fill in your funding and payment information. This will allow Wiley to send you a quote for the article processing charge (APC) in case of acceptance. This quote takes into account any reduction or fee waivers that you may be eligible for. Authors do not need to pay any fees before their manuscript is accepted and transferred to the publisher.

EMBO Press participates in many Publish and Read agreements that allow authors to publish Open Access with reduced/no publication charges. Check your eligibility: <https://authorservices.wiley.com/author-resources/Journal-Authors/open-access/affiliation-policies-payments/index.html>

***** PLEASE NOTE ***** As part of the EMBO Press transparent editorial process initiative (see our Editorial at <https://dx.doi.org/10.1038/msb.2010.72>), Molecular Systems Biology publishes online a Review Process File with each accepted manuscripts. This file will be published in conjunction with your paper and will include the anonymous referee reports, your

point-by-point response and all pertinent correspondence relating to the manuscript. If you do NOT want this File to be published, please inform the editorial office at msb@embo.org within 14 days upon receipt of the present letter.

Reviewer #1:

Sommerauer & Gallardo-Dodd et al report here the identification of ER regulated genes involved in its protective activities towards NAFLD. In particular, the authors identified a role for TEAD1 in activating lipogenic pathways.

Main concerns:

- 1) The authors treated male mice with ER agonists. However, ER has been reported to display different activities in male versus female mice. Several studies have reported a lack of effect of ER in male mouse livers with regards to protection against high fat diet induced steatosis (e.g. Meda et al Mol Metab 2020). Therefore, the choice to focus the entire study on males might be at odds with previous literature.
- 2) In this context, it remains unclear whether the findings relate to sex-related differences in liver pathophysiology and whether the identified targets including TEAD1 have any role in explaining sex differences in sensitivity to NAFLD.
- 3) Data from Fig.3 show that ER regulated genes are expressed mostly in hepatocytes in different species. This might result from the fact that ER regulated genes were defined using bulk RNAseq from whole livers. Although this does not impede identifying deregulated genes in non-hepatocytes, this might have nevertheless skewed detection of deregulated genes towards those expressed in the most prominent cell type. Additionally, gene expression changes were analyzed after long-term ER agonist treatment, which might not allow to best capture the primary and direct ER target genes.
- 4) Along the same line, the choice of the selected subset of ER reverted genes is based on differentially acetylated promoter/enhancers rather than on assessment of direct binding and regulation by ER (Fig5)
- 5) Fig5: was sex taken into account? Are there sex-related differences in ER regulated gene expression in this cohort?
- 6) The ER regulated gene signature was found to be associated with fibrosis in the human data. TEAD factors exert important roles in hepatic stellate cells. Supplementary Table 10 indicates that stromal cells express the highest levels of Tead1 in the mouse. Fibrotic livers are enriched for activated stellate cells or fibroblasts, which could therefore represent the main source of TEAD1 and explain the link with liver fibrosis observed (rather than Tead1 expression in hepatocytes).
- 7) Along the same, line, ER mediated control of TEAD1 is only reported in whole livers of mice with long-term agonist treatment, which does not rule out that the effect could stem from changes in fibroblast activation and/or TEAD1 expression. Direct evidence of ER regulating TEAD1 in hepatocytes is lacking.
- 8) The used inhibitors are pan-TEAD inhibitors. The authors would need to define which TEAD factors are expressed in their model systems in order to define whether this is consistent with a primary role for inhibition of TEAD1 in observed gene regulation and cellular metabolism.
- 9) Data in primary human hepatocytes (including RNAseq data) were obtained using cells from a single donor, which might compromise the generalization of the drawn conclusions.

Additional points:

- 1) Fig3D: how was the analysis performed? Are differences significant?
- 2) Page 7: "Activation of ER-responsive pathways is mediated through changes in chromatin accessibility" should be rephrased as accessibility per se was not assessed
- 3) Neither the number of replicates used for RNAseq and ChIPseq nor their consistency is indicated

Reviewer #2:

Summary

The submitted manuscript aims to dissect the role of estrogen receptor signaling in the development of nonalcoholic fatty liver disease. Using an in vivo mouse model, publicly available human gene expression, and primary human hepatocytes (PHH), the authors investigated how estrogen receptor (ER)-mediated signaling attenuated NAFLD mice developed from a high fat diet (HFD) and the translatability of the results to humans. The authors report that ER activation in HFD fed male mice attenuated NAFLD severity. Bulk and single-cell RNA seq identified effects on key liver pathways, extending beyond lipid metabolism. ChIP-seq for histone acetylation identified enhancers near Tead1 gene and that Tead1 was induced in HFD fed mice which reverted upon ER agonist treatment. The functional role of Tead1 in lipogenesis was further demonstrated in PHH spheroids using two antagonists and short interfering RNA that provided compelling supportive evidence. Reduction of hepatic steatosis in male mice following ER-agonist treatment is noteworthy suggesting estrogen may have a hepatoprotective effect. Additionally, the identification of ER-controlled gene networks with translational relevance to primates underscores the significance of these results. The study will be of interest to investigators and clinicians with interests in hepatology, endocrinology, and molecular biology as well as others developing treatments for NAFLD. Public health professionals concerned with the rising prevalence of NAFLD may also find this study valuable as it contributes to our understanding of potential therapeutic approaches.

Major points:

- Results: The liver and body weights are reported individually. However, visually there appears to be a correlation between the two. The data should also be examined as relative liver weight normalized to body weight to confirm that the liver weights truly change.
- Results: The histological assessment is superficial. While the lipid accumulation is evident in the photomicrographs, it is not clear how/whether other common features of NAFLD (Inflammation, fibrosis) were evaluated. This is important because the authors state that they did not see fibrosis which can be difficult to detect with H&E. Were the slides evaluated by a pathologist or someone experienced with liver histology to obtain either quantitative or qualitative scores? Were there notable spatial pathologies (e.g., periportal vs. pericentral steatosis)? The authors are encouraged to share additional images, digitized slides, or high-quality versions of the photomicrographs either through Figshare or Bioimage Archives.
- Results: The "HFD and ER activation signatures co-occur in the liver and are maintained between mouse and primates" section should be re-examined and revised. The authors claim that estrogen agonists and NAFLD similarly affect human and macaque cells. However, the single-cell data appears to only include normal liver, at least for the macaque. There is insufficient information about the public dataset study designs in the manuscript to adequately evaluate this claim. Further details regarding about how these data were used must be described. Some statements (e.g., "Our single-cell analysis confirmed ...") and the methods suggest these data were generated by this group but that data and most of the processing was done by the liver cell atlas group at VIB/Ghent University.
- Methods: The transcriptomic-based signal-to-noise ratio (tSNR) is poorly defined in the methods section (How is variance calculated? What is a "noise baseline"? Is tSNR calculated the same way as in the publication 10.1016/j.celrep.2020.107795 or another previous publication?).

Minor points:

- Methods: Provide additional details regarding the in vivo study design. What was the housing temperature? What approximate Zeitgeber time were samples collected and was this consistent? When during the study was blood glucose measured? How many times were the agonists injected? How was the dose selected?
- Results: The authors state that "[as] previously reported, estrogenic ligand treatments modestly reduced total weight, liver weight, and blood glucose levels" citing reference 11. Please report liver weight data from this study.
- Results: In the Results section, "Systemic ER activation mitigates diet-induced liver alterations in an isoform-specific manner", is vague. What does "across all male comparisons" mean. Please explicitly state that the 1,477 genes represents the union of all genes in the main text and make it more evident in the Figure 2B.
- Figure 1: Panels on fig 1 should be repositioned - the B and D panels are out of order.
- Figure 2: Please clarify that the gray scale from Figure 2A reflects the colors in the barplot Figure 2B. This is not indicated anywhere.
- Figure 2: Some of the numbers between the main text and figures appear to not match ("('non-reverted' (n = 333)" appears as 335 in Figure 2B. Reverted also seems incorrect.
- Figure 2: Why is there a smaller circle in Figure 2C (Positive regulation of gene expression). From the methods it seems that terms were collapsed and named after the parent term.
- Figure 3C: The authors state that "ER β -specific effects were enriched in the vasculature including capsule, portal, and central vein" while there appears no enrichment whatsoever in the hepatocytes. How was this determined?
- Figure 3D: Can the authors clarify what min - max scaling represents. Is the scaling done across all scores, only within a function, or only within a cell type?
- Figure 5: The use of the same color scale for z-score and a binary value of high and low is confusing. EV5 similarly uses the same color scale for two different values.
- All figures: Creating 4 different categories of figures (Figures, Appendix, EV Figures and Supplementary Figures) is confusing. Is this consistent with journal policies? It seems like both the Appendix and EV figures should be supplemental figures and the imaging dataset referenced in the methods as the imaging dataset with the Figshare link.
- The term ES-E-G is not defined in the main text.

Reviewer #3:

In this paper, Sommerauer, Gallardo-Dodd and colleagues have investigated the role of isoform-selective ER activation on NAFLD disease course, uncovering an important contribution of TEAD1. The paper is well-written, the methodology is sound and the conclusion of great interest given the lack of current treatment options for this prevalent disease.

I have the following remarks:

- The authors showed that the expression of estrogen-sensitive genes correlated with disease severity in a large NAFLD cohort. Was there a major difference in gene expression profile between male and female patients? Is the predictive capacity retained when splitting between men and women (also given that female patients might have had less severe fibrosis)?
- For ChIP-seq, why was ER α investigated using E2-treated instead of PPT-treated mice, given that E2 is non-specific and activates both ER isoforms, and that PPT had a stronger effect on gene expression in clusters 3 and 4 (Figure 2A)?
- The data on the therapeutic effect of TEAD1 would be made stronger by showing an in vivo effect. Given the regulation of

TEAD1 by estrogen, would a therapeutic effect in mice/humans depend on sex, or work equally well in males and females?

Minor points:

- The authors claim that "Treatment with ER β agonists may pose a future treatment strategy for diet-induced fibrosis" (p10) seems a bit premature given the lack of direct demonstration of an antifibrotic effect. In this context, the authors could cite PMID 28884481, in which the effect of ER agonists on CCl₄-induced liver fibrosis was investigated.
- Following a recent international consensus process, the nomenclature of NAFLD was proposed to change to MASLD - metabolic dysfunction-associated steatotic liver disease (Rinella M, A multi-society Delphi consensus statement on new fatty liver disease nomenclature). I would suggest adapting the terminology in this paper.
- The TEAD inhibition-induced changes in oxygen consumption (Figure 5G) seem quite minor. Although statistically significant, are these relevant?
- In view of Figure 3, I do not see the added value of the analyses in Figure 2C.
- The authors mention that "As previously reported, estrogenic ligand treatments modestly reduced total weight, liver weight and blood glucose levels" (page 5). These changes, especially for glucose, are very small. I would refrain from such statements when the difference is not statistically significant.

We thank the reviewers for their assessments of our work and their constructive comments. This has helped us design new analysis and experiments to improve the clarity and the impact of our findings to the field, while adhering to the journal's requirements. For the reviewers' convenience, we provide the original comments in black and our point-by-point responses in red. Amendments in the main text are also highlighted in red. We named figures that were generated in response to the reviewers as "Fig RX", while we refer to manuscript figures as they appear in the main manuscript. We trust that all comments have been addressed adequately either through additional analysis/experiments or through clarifying specific points.

We have renamed 'nonalcoholic fatty liver disease (NAFLD)' to 'metabolic dysfunction-associated steatotic liver disease (MASLD)' in our response to the reviewers and manuscript to adopt non-stigmatising nomenclature as recently proposed (Rinella et al. 2023b, 2023a, 2024) and as suggested by reviewer 3.

Reviewer #1:

Sommerauer & Gallardo-Dodd et al report here the identification of ER regulated genes involved in its protective activities towards NAFLD. In particular, the authors identified a role for TEAD1 in activating lipogenic pathways.

Main concerns:

1) The authors treated male mice with ER agonists. However, ER has been reported to display different activities in male versus female mice. Several studies have reported a lack of effect of ER in male mouse livers with regards to protection against high fat diet induced steatosis (e.g. Meda et al Mol Metab 2020). Therefore, the choice to focus the entire study on males might be at odds with previous literature.

We thank the reviewer for pointing out that previous studies have shown sex-differences of ER function in hepatocytes, such as Meda et al. In these studies, ER genes were genetically modified in mice, while our approach involved the injection of ER agonist. Thus, the observation by Meda et al. does not necessarily extend to extra-hepatic ERs, or ERs expressed in other liver cell types (e.g., hepatic stellate cells or immune cells), which may mediate protective effects.

In contrast, other studies have shown that systemic estrogen signaling in male mice can convey protective effects. For example, the ablation of endogenous E2 through aromatase knockout renders male mice susceptible to metabolic abnormalities (Hewitt et al. 2004), a phenotype that can be rescued through E2 supplementation. These findings emphasize the connection between estrogen signaling and hepatic safeguarding in males. Consequently, we believe our findings complement current literature.

2) In this context, it remains unclear whether the findings relate to sex-related differences in liver pathophysiology and whether the identified targets including TEAD1 have any role in explaining sex differences in sensitivity to NAFLD.

Indeed, our study focused on estrogen signaling in male mice, as the high-fat diet fed females were protected from hepatic steatosis even without ER agonist treatment. Drawing conclusive evidence on whether the identified ER-sensitive target genes contribute to sex-related NAFLD/MASLD susceptibility is challenging, as it would require individual knockouts of every ER-sensitive gene in both sexes, which may be detrimental. For instance, *Tead1* deletion is embryonically lethal, further hampering experimental evidence (Chen et al. 1994).

To assess whether the ER-sensitive gene expression changes in females as well, we plotted the mRNA levels of the ER-sensitive genes in female and male mice of both sexes and diets by adding an additional panel to the heatmap in Fig 5A. We found that most ER-sensitive genes identified in male mice remained unchanged upon HFD in females. While this is no definite evidence explaining sexual dimorphism of MASLD prevalence, it suggests that these genes are influenced by ER signaling in females as well.

Furthermore, *Tead1* is sex-specifically expressed in control diet conditions; however, its expression increases upon HFD in male mice to a similar extent as in female mice (**Fig. R1**).

Fig. R1. Tead1 gene expression is sex-specific.

Lineplot shows TPM-normalized *Tead1* expression levels in female (f) and male (m) mice on control (CD) and high fat diet (HFD) and male ER agonist-treated HFD-fed mice.

3) Data from Fig.3 show that ER regulated genes are expressed mostly in hepatocytes in different species. This might result from the fact that ER regulated genes were defined using bulk RNAseq from whole livers. Although this does not impede identifying deregulated genes in non-hepatocytes, this might have nevertheless skewed detection of deregulated genes towards those expressed in the most prominent cell type. Additionally, gene expression changes were analyzed after long-term ER agonist treatment, which might not allow to best capture the primary and direct ER target genes.

As noted by the reviewer, the liver predominately consists of hepatocytes, allowing for a more accurate determination of gene expression changes through our bulk RNA-seq dataset. Therefore, an enrichment for hepatocyte genes in our differential expression analysis is expected. However, our analysis also successfully detected significant changes for genes primarily expressed in non-hepatocyte cells. For example, our examination of gene signatures in public single-cell data revealed that genes with an ER β -specific response (n=239) were highly expressed in endothelial and stromal cell types. Furthermore, many of the reverted genes are lipid metabolic genes, which are known to be mainly expressed in hepatocytes. Consequently, the mapping of gene expression using single cell data aligns with the known primary locations of cellular pathways and processes.

We acknowledge the reviewer's observation that three weeks of estrogen treatment may mask immediate primary effects upon ER stimulation. While a time-series of ER agonist treatments would resolve the dynamics of protection/reversal of HFD-induced changes, it would necessitate a large number of mice. We opted for a three week duration of estrogen treatment to account for the necessary time for alleviating of the HFD-induced phenotype. Nevertheless, we would like to point out that due to the systemic effects of ER activation, a mixture of direct and secondary effects is to be anticipated regardless of the time point.

In response to these concerns, we added the following sentence into the limitation section:

“Our study investigated only a single time point after three weeks of ER agonist treatment. Consequently, the identified ER-sensitive genes are likely a combination of direct and indirect effects of ER signaling. Moreover, the detection of gene signatures relied on transcriptomic differences from bulk liver samples, potentially overlooking subtle changes in low-abundant cell types.”

4) Along the same line, the choice of the selected subset of ER reverted genes is based on differentially acetylated promoter/enhancers rather than on assessment of direct binding and regulation by ER (Fig5).

The identified estrogen-sensitive genes are not strictly directly regulated by ER on the chromatin. Hence, we renamed these genes from ‘ER-sensitive / estrogen-sensitive’ to ‘ER-regulated / estrogen-regulated’ to avoid potential misunderstandings. Moreover, the reviewer is correct in noting that the selection of ER-sensitive genes was not based on ER-binding but rather on the sensitivity of enhancers towards ER activation. Despite this, our analysis of public ER α ChIP-seq data indicated the presence of several ER α binding sites in and near *Tead1* and *Acot* genes (Fig 4B, Fig EV4F).

5) Fig5: was sex taken into account? Are there sex-related differences in ER regulated gene expression in this cohort?

We understand the reviewer’s concern. Unfortunately, the cohort used from Govaere et al. (Govaere et al. 2020) lacks metadata on patient sex and age, which we were unable to obtain despite our request to the authors. To address this issue, we annotated RNA-sequenced patient samples by sex using expression information from sex-chromosome marker genes (i.e., *XIST*, *DDX3Y*). In Fig. R2, we present a heatmap of ER-regulated genes, similar to Fig. 5A, where the RNA-seq samples are separated by sex. The expression profiles of these genes are overall consistent for both male and female groups, suggesting no major sex differences.

We would like to note that many of these ER-regulated genes remain unchanged in female mice on HFD compared to controls. This observation may be explained by the fact that many female patients are likely post-menopausal (average age reported by the authors is 54 years), aligning with the absence of protective effects mediated by ER signaling in MASLD.

Fig. R2. ER-regulated genes in mouse are not sex-specifically regulated in humans. Heatmaps display changes in expression levels for the 45 orthologous ES-E-G genes in MASLD/NAFLD patients and mice separated by sex (male: left; female: right). Color gradient indicates z-score-normalized gene expression counts in males (blue: green and red: brown) and in females (blue: purple and red: brown). Four k-means clusters group genes by expression in healthy (CTRL), NAFL and NASH patients, as well as patients with different NAS (early (E): NAS0-1, moderate (M): NAS2-6, advanced (A): NAS7-8) and fibrosis stages (E: F0-1, M: F2, A: F3-4). Expression levels of the 45 genes in HFDm and HFDf are shown. Color code distinguishes downregulated and upregulated genes in HFDm versus CDM. Gene names follow human nomenclature.

6) The ER regulated gene signature was found to be associated with fibrosis in the human data. TEAD factors exert important roles in hepatic stellate cells. Supplementary Table 10 indicates that stromal cells express the highest levels of Tead1 in the mouse. Fibrotic livers are enriched for activated stellate cells or fibroblasts, which could therefore represent the main source of TEAD1 and explain the link with liver fibrosis observed (rather than Tead1 expression in hepatocytes).

We appreciate this observation by the reviewer. However, the RNA-seq data from this cohort did not indicate a strong correlation between *TEAD1* gene expression and fibrosis (**Fig. R3A**). In fact, *TEAD1* appeared to increase in early stages (F1) and decrease at advanced stages (F3-4). Moreover, by inspecting the contribution of *TEAD1* to our prediction models for MASLD/NAFLD stage, NAS and fibrosis (**Fig. R3B**), we observe that *TEAD1* gene expression is informative for early MASLD/NAFLD stages and NAS values, but not fibrosis. Despite this, other genes from the ER-regulated set show stronger association with fibrosis (e.g., *MOXD1*). We also agree with the reviewer that the higher expression of *Tead1* in murine stromal cells could be indicative of roles in fibrosis, although this aspect is not described in this study and warrants further investigation.

Fig. R3. *TEAD1* gene expression is indicative for MASLD/NAFLD stage and NAS, but not fibrosis

(A) Box plot shows CPM-normalized *TEAD1* gene expression across fibrosis stages in the MASLD/NAFLD patient cohort. Boxes cover from the first to the third quartile and whiskers extend from the hinges to the minimum and maximum values up to $1.5 \times \text{IQR}$. (B) Lollipop plots demonstrate the importance of *TEAD1* in model classifications of MASLD/NAFLD stage, NAS and fibrosis (caret R package).

7) Along the same, line, ER mediated control of TEAD1 is only reported in whole livers of mice with long-term agonist treatment, which does not rule out that the effect could stem from changes in fibroblast activation and/or TEAD1 expression. Direct evidence of ER regulating TEAD1 in hepatocytes is lacking.

To investigate whether ER α could regulate *TEAD1* gene expression in human hepatocytes, we examined publicly available ER α -ChIP-Seq data from primary human hepatocytes (from a male and a female donor, pooled prior to the ChIP-experiment (Collins et al. 2021)). We found enrichment of ER binding within the *TEAD1* gene (Fig. R4A), but not in the *TEAD2*, *TEAD3* or *TEAD4* genes (Fig. R4B-D), suggesting that ER α could directly regulate *TEAD1* expression in hepatocytes.

Of note, while this finding suggests partial direct ER α involvement in hepatocytes, it does not exclude the possibility that signals from other cell types, such as immune cells or hepatic stellate cells, contribute to *TEAD1* gene expression in hepatocytes through secondary signaling pathways. Our functional analysis in liver spheroids (Fig 5) demonstrated that *TEAD* inhibition reduces steatosis in hepatocytes. However, our data do not indicate that *TEAD1* is involved in development of fibrosis or that *TEAD* inhibition reduces fibrosis.

Fig R4. ER α binds to *TEAD1*, but not *TEAD2*, *TEAD3* or *TEAD4*.

(A-D) Genome browser views (IGV, hg38) show genomic regions around all four *TEAD* gene loci. Genomic locations and sizes are indicated. The y-axis of each track specifies

normalized ChIP-seq read density (parenthesis) of ER α (blue) or input (red) (GSE158856). Black boxes represent exons and UTRs, connecting lines indicate intronic sequences. Arrows indicate directionality of gene transcription.

8) The used inhibitors are pan-TEAD inhibitors. The authors would need to define which TEAD factors are expressed in their model systems in order to define whether this is consistent with a primary role for inhibition of TEAD1 in observed gene regulation and cellular metabolism.

As shown in Dataset EV8, TEADap inhibits TEAD1, TEAD2 and TEAD3 to a similar extent, while TEADsf inhibits all four TEAD isoforms (TEAD1-TEAD4), also to a similar extent. Consequently, the effects observed in the primary hepatocytes could result from the inhibition of any TEAD protein or a combination of them. However, our expression data in Dataset EV8 indicates that only *Tead1* gene expression levels are significantly changed by HFD and ER-agonist treatments in mouse liver. Moreover, *TEAD1* gene expression is the highest in human samples and is increased throughout MASLD progression.

9) Data in primary human hepatocytes (including RNAseq data) were obtained using cells from a single donor, which might compromise the generalization of the drawn conclusions. In response, we have included an additional paragraph in the discussion section of the revised manuscript in which we acknowledge that the sex of hepatocyte donors might impact the presented results:

"Inter-individual variability, including demographic, environmental and genetic factors can impact outcomes when working with human primary cells. However, we previously did not observe major differences in the molecular effects of YAP/TEAD inhibitors [40] despite MASH phenotypes slightly varied when growing spheroids derived from different donors. While these findings argue against major differences of the molecular networks underlying YAP/TEAD inhibition, a modulating role of various individual factors cannot be excluded."

Additional points:

1) Fig3D: how was the analysis performed? Are differences significant?

To clarify the analysis for the reader, we have extended the relevant part in the methods section: "Enrichment scores for the relevant ER activation signature gene sets and Reactome pathway clusters identified were calculated using pagoda2 (v1.0.2). Up to 5,000 cells for each annotated cell type were subsampled for the analysis. Pathway activity scores were aggregated at the cell type level by averaging the enrichment values of all individual cells annotated for a given cell type cluster and condition. To ensure comparability of pathway activity scores, the scores were scaled to a 0-1 range using the min-max scaling method across all cell types for each pathway. Changes in pathway activity were measured as the difference between control and HFD scores for each cell type".

Unfortunately, the implementation of this method in the pagoda2 package does not include a statistical approach to assess significant differences in pathway activities.

2) Page 7: "Activation of ER-responsive pathways is mediated through changes in chromatin accessibility" should be rephrased as accessibility per se was not assessed

We agree that while H3K27ac is commonly regarded as a marker for accessible chromatin, it does not inherently prove chromatin accessibility. We have revised the section title to: "Activation of ER-responsive pathways is mediated through chromatin changes".

3) Neither the number of replicates used for RNAseq and ChIPseq nor their consistency is indicated.

We added the number of replicates for the RNA-seq (n=4 mice per condition) and ChIP-seq (n=3 mice per condition) experiments in the figure legends and methods.

- RNA-seq:

Methods:

“RNA sequencing and data processing. Strand-specific RNA libraries (**n=4 mice per condition**) were generated using the NEBNext Ultra II stranded library kit (New England Biolabs) combined with polyA-coupled beads (New England Biolabs) according to the manufacturer’s instructions.”

Figure legends:

Fig 1A: “Schematic representation of the mouse experimentation. Five-week-old female (f) and male (m) C57BL/6 mice (**n=4**) received either...”

Fig 2A: “... unified deregulated genes (DEGs, n=1,477) in mice on different diets and ER-agonist treatments (color-coded, **n=4**).”

Fig EV2A-B: “Four-way Venn diagrams show intersections and numbers (n) of gene sets that are either (A) upregulated (red arrow) or (B) downregulated (blue arrow) by ER-agonist treatments compared to HFD in male mice (**n=4 mice per condition**)”.

- ChIP-Seq:

Methods:

“ChIP-sequencing and data analysis. Formaldehyde-fixed livers (**n=3 mice per condition**) were homogenized using a douncer and washed twice with ice-cold PBS.”

Figure legends:

Fig 4A: “The average signal is depicted (**n=3 mice per condition**).”

Fig EV4C: “Box plots illustrate number of normalized (1× genome coverage) reads in peaks (log₂) for the same regions and diet comparisons as in **Fig 4A (n=3 mice per condition)**.”

Reviewer #2:

Summary

The submitted manuscript aims to dissect the role of estrogen receptor signaling in the development of nonalcoholic fatty liver disease. Using an in vivo mouse model, publicly available human gene expression, and primary human hepatocytes (PHH), the authors investigated how estrogen receptor (ER)-mediated signaling attenuated NAFLD mice developed from a high fat diet (HFD) and the translatability of the results to humans. The authors report that ER activation in HFD fed male mice attenuated NAFLD severity. Bulk and single-cell RNA seq identified effects on key liver pathways, extending beyond lipid metabolism. ChIP-seq for histone acetylation identified enhancers near Tead1 gene and that Tead1 was induced in HFD fed mice which reverted upon ER agonist treatment. The functional role of Tead1 in lipogenesis was further demonstrated in PHH spheroids using two antagonists and short interfering RNA that provided compelling supportive evidence. Reduction of hepatic steatosis in male mice following ER-agonist treatment is noteworthy suggesting estrogen may have a hepatoprotective effect. Additionally, the identification of ER-controlled gene networks with translational relevance to primates underscores the

significance of these results. The study will be of interest to investigators and clinicians with interests in hepatology, endocrinology, and molecular biology as well as others developing treatments for NAFLD. Public health professionals concerned with the rising prevalence of NAFLD may also find this study valuable as it contributes to our understanding of potential therapeutic approaches.

Major points:

- Results: The liver and body weights are reported individually. However, visually there appears to be a correlation between the two. The data should also be examined as relative liver weight normalized to body weight to confirm that the liver weights truly change.

As the reviewer pointed out, it appears that liver weight and body weight correlate. By plotting the ratios between both weight measurements (Fig. R5), we found almost no change upon HFD in male mice, and this ratio remains unchanged in the agonist-treated HFD fed males. Under ‘Results’, we stated liver weight increased, but do not specify that the increase is disproportionately to body weight.

Fig R5. Liver and body weight ratios are unchanged.

Bar plot shows the liver weight relative to body weight in female (fe) and male (ma) mice fed a control diet (CD) or a high fat diet (HFD) as well as male HFD mice treated with ER agonists (color-coded). Dots indicate individual biological replicates.

- Results: The histological assessment is superficial. While the lipid accumulation is evident in the photomicrographs, it is not clear how/whether other common features of NAFLD (Inflammation, fibrosis) were evaluated. This is important because the authors state that they did not see fibrosis which can be difficult to detect with H&E. Were the slides evaluated by a pathologist or someone experienced with liver histology to obtain either quantitative or qualitative scores? Were there notable spatial pathologies (e.g., periportal vs. pericentral steatosis)? The authors are encouraged to share additional images, digitized slides, or high-quality versions of the photomicrographs either through Figshare or Bioimage Archives.

We did not assess inflammation, fibrosis, or spatial steatosis by histological staining.

However, studies investigating mice on a comparable HFD regimen for a similar time strongly suggests that liver inflammation occurs (Pilling et al. 2021). Reporting low fibrosis level without the presence of histological assessment might be confusing. However, we refer to our transcriptomic data that indicated little change of fibrosis-associated marker genes. To avoid confusions, we modified the sentence in the discussion section:

From: “Although our HFD model did not induce fibrosis...” to: “**While the expression of fibrosis-associated genes was unchanged in our HFD model**, the ER β agonists specifically and predominantly suppressed a range of genes associated with the extracellular matrix, angiogenesis and growth factor signaling”.

As recommended by the reviewer, we uploaded the original H&E stainings to Figshare.

- Results: The "HFD and ER activation signatures co-occur in the liver and are maintained between mouse and primates" section should be re-examined and revised. The authors claim that estrogen agonists and NAFLD similarly affect human and macaque cells. However, the single-cell data appears to only include normal liver, at least for the macaque. There is insufficient information about the public dataset study designs in the manuscript to adequately evaluate this claim. Further details regarding about how these data were used must be described. Some statements (e.g., "Our single-cell analysis confirmed ...") and the methods suggest these data were generated by this group but that data and most of the processing was done by the liver cell atlas group at VIB/Ghent University.

We appreciate the reviewer's concern, as it is true that the macaque dataset lacks a MASLD condition. To address potential confusion, we have revised the title of this section to "***HFD and ER activation signatures co-occur in the liver across species***". We are grateful to the Liver Cell Atlas group for providing this invaluable resource, and fully acknowledged the importance of clarify in utilizing public resource. The following changes have been incorporated (in bold):

- Results:
 - "[...], we analyzed **public** single-cell (comprising 483,955 cells) and spatial transcriptomics datasets".
 - "The same cell types were **enriched** when mapping these gene signatures to reference human and **healthy** macaque single-cell liver atlases. This suggested that the hepatic molecular key signatures and the cellular architecture altered by HFD or in MASLD **affect similar cell types** in mice and humans, and that the observed gene regulatory responses to estrogen treatment **are partly shared**".
 - "We find that these gene signatures are **in part shared between mouse and human**, and that systemic ER activation protects the liver by counteracting these changes".
- Discussion:

"**The analysis of single-cell data** confirmed that HFD induces inflammatory signaling and alters hepatic immune cell composition, potentially amplifying the responsiveness to or effects by estrogens due to increased proportions of immune cells".
- Methods:

"**Preprocessed** public single-cell and spatial transcriptomics datasets and annotations were retrieved from the Liver Cell Atlas. **Given our gene signatures were defined in male mice**, only cells originating from male mice samples were used in the analysis and primary cells were removed. **Accordingly, only cells** obtained from male macaque and human were considered. Cell type composition analyses were conducted in R using Seurat (v4.0.2). Enrichment scores for the relevant ER activation signature gene sets and Reactome pathway clusters identified were calculated using pagoda2 (v1.0.2). **Up to 5,000 cells for each annotated cell type were subsampled for the analysis. Pathway activity scores were aggregated at the cell type level by averaging the enrichment values of all individual cells annotated for a given cell type cluster and condition. To make pathway activity scores comparable, the scores were scaled to a 0-1 range using the min-max scaling method across all cell types for each pathway. Changes in pathway activity were measured as the difference between control and HFD scores for each cell type.**"

- Methods: The transcriptomic-based signal-to-noise ratio (tSNR) is poorly defined in the methods section (How is variance calculated? What is a "noise baseline"? Is tSNR calculated

the same way as in the publication 10.1016/j.celrep.2020.107795 or another previous publication?).

The transcriptomic-based signal-to-noise ratio (tSNR) measure was calculated between our groups of interest as described in the publication (Lopes-Ramos et al. 2020) mentioned by the reviewer. We have changed the relevant section in the methods and added the citation: “Transcriptome-wide differences across conditions were measured unbiased by using a transcriptome-based signal-to-noise ratio (tSNR) as described previously [49]. For this, the Euclidean metric was used as a measure of distance across transcriptomes. The signal was defined as the distance between the averaged transcriptomes of two groups while the noise was defined based on the total within-group variation observed (i.e., the dispersion of distance measurements of each sample transcriptome to the group average), expressed as: [...]”.

Minor points:

- Methods: Provide additional details regarding the in vivo study design. What was the housing temperature? What approximate Zeitgeber time were samples collected and was this consistent? When during the study was blood glucose measured? How many times were the agonists injected? How was the dose selected?

We clarified and expanded the method section (in bold):

Animal experiments and tissue preparation. Animal experimentation has been previously reported **and ARRIVE guidelines were followed** [11]. In short, five- to six-week-old male and female C57BL/6J mice obtained from in-house breeding were fed a control (D12450J, 10% kcal fat, Research Diet) or high-fat diet (D12492, 60% kcal fat, Research Diet) *ad libitum* for 13 weeks (**n=4 per condition**). Subsets of male mice on HFD were additionally injected intraperitoneally with the estrogenic ligands 17 β -estradiol (E2, 0.5mg/kg body weight, Sigma-Aldrich), 4,4',4''-(4-Propyl-[1H]-pyrazole-1,3,5-triyl)trisphenol (PPT, 2.5mg/kg body weight, Tocris), 2,3-Bis(4-hydroxyphenyl)propionitrile (DPN, 5 mg/kg body weight, Tocris) and 4-(2-(3,5-dimethylisoxazol-4-yl)-1H-indol-3-yl)phenol (DIP, 10mg/kg body weight) or given a sham injection every second day from week 10 to week 13 (**n=4 per condition**). **Ligand concentrations were chosen according to literature [13,41,42]**. The ligands were diluted in 55% water, 40% PEG400 and 5% DMSO. Mice in each group were descended from different parents and were housed in at least two different cages **at 20°C and sacrificed at Zeitgeber time 3 to 4**. Upon sacrifice, **blood glucose was measured after 2h fasting with a glucometer (Accu-Chek) and livers of C57BL/6J mice were dissected and washed with phosphate-buffered saline (PBS)**. Livers were either cross-linked for ChIP-seq, embedded for histology or flash-frozen in liquid nitrogen for RNA-seq.

- Results: The authors state that "[as] previously reported, estrogenic ligand treatments modestly reduced total weight, liver weight, and blood glucose levels" citing reference 11. Please report liver weight data from this study.

This point was also brought up by reviewer 3. We changed the sentence to:

“Liver weight and blood glucose levels did not exhibit significant changes with any estrogenic ligand treatments, and total weight was significantly decreased upon DPN treatment (**Fig EV1, A-C**) [11].”

- Results: In the Results section, "Systemic ER activation mitigates diet-induced liver alterations in an isoform-specific manner", is vague.

We changed the result subheader to: “Systemic activation of ER α and ER β mitigate diet-induced gene signatures”

What does "across all male comparisons" mean.

Please explicitly state that the 1,477 genes represents the union of all genes in the main text and make it more evident in the Figure 2B.

We clarified the sentence: “We **formed the union of DEGs across the five male comparisons (CDm vs HFDm, HFDm vs DPN/DIP/E2/PPT, n=1,477), which separated into four distinct expression clusters (Fig 2A)**”.

- Figure 1: Panels on fig 1 should be repositioned - the B and D panels are out of order.

We have revised the order in which the panels are mentioned in the text.

- Figure 2: Please clarify that the gray scale from Figure 2A reflects the colors in the barplot Figure 2B. This is not indicated anywhere.

We added a panel above the horizontal bar chart in Fig 2B to illustrate that the colors are indicative of cluster membership and modified the figure legend to include this information: “[...] Horizontal bar chart displays the proportional occurrences of gene sets in the four clusters (**gray scale as in Fig 2A**)”.

- Figure 2: Some of the numbers between the main text and figures appear to not match ("non-reverted" (n = 333)" appears as 335 in Figure 2B. Reverted also seems incorrect.

We have corrected the numbers in the main text to “non-reverted (n=335)” and “reverted (n=379)”.

- Figure 2: Why is there a smaller circle in Figure 2C (Positive regulation of gene expression). From the methods it seems that terms were collapsed and named after the parent term.

As indicated by the reviewer, overrepresented GO terms were collapsed into groups based on their semantic similarity using the *rrvgo* R package. This approach uses a similarity threshold to group GO terms. In this case, the selected cutoff was not sufficient to group the terms in “Positive regulation of gene expression” and “Positive regulation of ERK1 and ERK2 cascade”. Increasing this threshold has the limitation of collapsing other GO terms into broader categories that could be of less relevance to the reader. We realized that the similarity threshold mentioned in the methods should have been 0.9 instead of 0.8, and we have made the correction.

- Figure 3C: The authors state that "ERβ-specific effects were enriched in the vasculature including capsule, portal, and central vein" while there appears no enrichment whatsoever in the hepatocytes. How was this determined?

The reviewer is correct in noting that hepatocytes do not exhibit a high enrichment for the ERβ-specific gene signature. As shown in the single-cell data in Fig 3B, the highest enrichment of this signature corresponds to endothelial and stromal cells. Additionally, from the spatial transcriptomics data in Fig 3C, it is evident that the ERβ-specific signature is predominately enriched in the peri-portal area (darker points), and to a lesser extent at the capsule and central vein. This enrichment is not necessarily originating from hepatocytes but rather other cell types that demonstrated a greater enrichment for this signature.

- Figure 3D: Can the authors clarify what min - max scaling represents. Is the scaling done across all scores, only within a function, or only within a cell type?

We have included the following sentence in the methods to clarify this scaling step:

“To make pathway activity scores comparable, the scores were scaled to a 0-1 range using the min-max scaling method across all cell types for each pathway”.

- Figure 5: The use of the same color scale for z-score and a binary value of high and low is confusing. EV5 similarly uses the same color scale for two different values.

We amended the color coding in both Fig 5 and Fig EV5.

- All figures: Creating 4 different categories of figures (Figures, Appendix, EV Figures and Supplementary Figures) is confusing. Is this consistent with journal policies? It seems like both the Appendix and EV figures should be supplemental figures and the imaging dataset referenced in the methods as the imaging dataset with the Figshare link.

We changed Appendix to “Appendix Fig S[number]” and “Supplementary Table” to “Dataset EV” or “Table EV” according to journal policy. We noticed that we wrongly referred to “Fig EV3C” as “Supplementary Figure 3C” in the Figure legend of Fig EV3 and corrected the mistake.

- The term ES-E-G is not defined in the main text.

We removed the previous ES-E-G definition, and we wrote out the term in the following paragraph:

“Among the 80 estrogen-sensitive enhancer gene pairs (ES-E-Gs), four enhancers [...]”.

Reviewer #3:

In this paper, Sommerauer, Gallardo-Dodd and colleagues have investigated the role of isoform-selective ER activation on NAFLD disease course, uncovering an important contribution of TEAD1. The paper is well-written, the methodology is sound and the conclusion of great interest given the lack of current treatment options for this prevalent disease.

I have the following remarks:

- The authors showed that the expression of estrogen-sensitive genes correlated with disease severity in a large NAFLD cohort. Was there a major difference in gene expression profile between male and female patients? Is the predictive capacity retained when splitting between men and women (also given that female patients might have had less severe fibrosis)?

We agree that this is an important point to consider, especially considering that ER-sensitive genes were identified using a male mouse model. Unfortunately, the cohort used from Govaere et al. (Govaere et al. 2020) lacks key metadata for this analysis, which we could not obtain from the authors despite our request. However, after imputing patient sex using expression of sex-chromosome marker genes, we found no major differences in ER-regulated gene expression between male and female MASLD patients (see Fig. R2 and response to reviewer #1).

In response to the reviewer’s subsequent question regarding ChIPseq, we included DIP and PPT-treated HFD samples for H3K27ac ChIP-seq and reanalyzed all ChIPseq data (please see detailed answer below). This resulted in a slightly different ES-E-G gene set, leading to poorer model performance. We removed former Fig 5B from the manuscript due to the lower relevance for the readership.

- For ChIP-seq, why was ER α investigated using E2-treated instead of PPT-treated mice,

given that E2 is non-specific and activates both ER isoforms, and that PPT had a stronger effect on gene expression in clusters 3 and 4 (Figure 2A)?

We initially included E2 due to its significant overlap with PPT and its widespread use in many studies, which may render it more relevant to the research community. However, to ensure that the observed effects on H3K27ac were not exclusively mediated through ER β , we conducted additional ChIP-seq experiments by incorporating the conditions HFDm+PPT and HFDm+DIP. Furthermore, we added a third replicate to all conditions to improve the statistical robustness of gene expression to enhancer H3K27ac correlations. This also allowed us to filter ES-E-G genes by applying a p-value cutoff ($p < 0.01$). We re-processed and re-analyzed the previous and newly obtained data using our prior approach.

Overall, the results remained very similar, encompassing (a) promoter and enhancer identification (Fig EV4A, EV4B), (b) the reversal of H3K27ac changes induced by HFD through ER agonists (Fig 4A, Fig 4B, Fig EV4C, EV4F) and (c) the identification of enhancer-gene pairs (Fig EV4C, EV4D).

We replaced all plots in Fig 4 and Fig EV4 with the newly analyzed data.

The downstream analysis yielded slightly slightly different gene sets. Specifically, we found a total of 49 genes (instead of 45 genes) as part of ES-E-Gs, with 33 genes overlapped with the original 45 identified genes (73%). Further, we added Appendix Fig S3, illustrating all 80 ES-E-G correlations. We updated all numbers in the main text and in fig EV4C.

Due to the change of the ES-E-G gene set, the analysis in human MASLD cohort also underwent slight modifications, and corresponding plots were updated (see response to point above).

Apart from updating the numbers in the manuscript, we changed the following sections:

Results:

“We identified 12,598 promoters and 26,210 enhancers, of which 142 promoters and 2,181 enhancers were differentially acetylated (DAc) at H3K27 upon HFD (**Fig EV4, A and B**). Most enhancer sites gained H3K27ac in response to HFD (69%), while promoter sites equally gained and lost H3K27ac (**Fig 4A**). We found that H3K27ac at both promoters and enhancer were partly restored by all ER and ER agonists (**Fig 4A and Fig EV4C**).”

Result subheader:

“Expression trends of ES-E-G genes follow MASLD disease progression in humans”

Furthermore, we updated data on ArrayExpress and scripts on GitHub.

- The data on the therapeutic effect of TEAD1 would be made stronger by showing an in vivo effect. Given the regulation of TEAD1 by estrogen, would a therapeutic effect in mice/humans depend on sex, or work equally well in males and females?

We have previously evaluated the effect of TEAD inhibitors in primary human hepatocyte cultures established from male and female donors (Oliva-Vilarnau et al. 2023). The results show that the canonical YAP/TEAD targets CTGF and CYR1 are effectively inhibited by both TEADap and TEADsf in both sexes (**Fig. R6**). Furthermore, we did not observe apparent differences at the transcriptomic level between the effect of TEADap and TEADsf (referred to as CMPD-3 and CMPD-4 in the aforementioned reference) on male and female hepatocytes.

Fig R6: TEADap and TEADsf inhibit YAP/TEAD target genes in hepatocytes of both sexes. Bar plots display relative gene expression levels of *CTGF* (left) and *CYR61* (right) upon treatment with increasing concentrations of TEADap and TEADsf inhibitors when normalized to DMSO treated vehicle controls. Data is presented for two female and one male donors.

Minor points:

- The authors claim that "Treatment with ER β agonists may pose a future treatment strategy for diet-induced fibrosis" (p10) seems a bit premature given the lack of direct demonstration of an antifibrotic effect. In this context, the authors could cite PMID 28884481, in which the effect of ER agonists on CCl4-induced liver fibrosis was investigated.

We included the mentioned reference.

- Following a recent international consensus process, the nomenclature of NAFLD was proposed to change to MASLD - metabolic dysfunction-associated steatotic liver disease (Rinella M, A multi-society Delphi consensus statement on new fatty liver disease nomenclature). I would suggest adapting the terminology in this paper.

We changed 'nonalcoholic fatty liver disease, NAFLD' to 'metabolic dysfunction-associated steatotic liver disease, MASLD'.

- The TEAD inhibition-induced changes in oxygen consumption (Figure 5G) seem quite minor. Although statistically significant, are these relevant?

We agree that the former Fig 5G shows a modest effect and its interpretation requires further assessment. Thus, we moved the former Fig 5G to the supplements (Fig EV5A).

- In view of Figure 3, I do not see the added value of the analyses in Figure 2C.

We acknowledge the similarity between analyses presented in Fig 3 and Fig 2C. However, Fig 2 introduces four distinct gene sets (reverted, non-reverted, DPN/DIP-specific and E2/PPT-specific) whose enrichments are not directly shown in Fig 3. Since these four gene sets are used throughout the manuscript, we believe that Fig 2 is an important result that should be introduced in the main text.

- The authors mention that "As previously reported, estrogenic ligand treatments modestly reduced total weight, liver weight and blood glucose levels" (page 5). These changes, especially for glucose, are very small. I would refrain from such statements when the difference is not statistically significant.

This point was also brought up by reviewer 2. We changed the sentence to:

“Liver weight and blood glucose levels did not exhibit significant changes with any estrogenic ligand treatments, and total weight was significantly decreased upon DPN treatment (**Fig EV1, A-C**) [11].”

References

- Chen Z, Friedrich GA, Soriano P. Transcriptional enhancer factor 1 disruption by a retroviral gene trap leads to heart defects and embryonic lethality in mice. *Genes Dev.* 1994
- Collins JM, Huo Z, Wang D. ESR1 ChIP-Seq Identifies Distinct Ligand-Free ESR1 Genomic Binding Sites in Human Hepatocytes and Liver Tissue. *International Journal of Molecular Sciences* 2021
- Govaere O, Cockell S, Tiniakos D, Queen R, Younes R, Vacca M, et al. Transcriptomic profiling across the nonalcoholic fatty liver disease spectrum reveals gene signatures for steatohepatitis and fibrosis. *Sci Transl Med* 2020
- Hewitt KN, Pratis K, Jones MEE, Simpson ER. Estrogen Replacement Reverses the Hepatic Steatosis Phenotype in the Male Aromatase Knockout Mouse. *Endocrinology* 2004
- Lopes-Ramos CM, Chen CY, Kuijjer ML, Paulson JN, Sonawane AR, Fagny M, et al. Sex Differences in Gene Expression and Regulatory Networks across 29 Human Tissues. *Cell Rep* 2020
- Oliva-Vilarnau N, Vorrink SU, Büttner FA, Heinrich T, Sensbach J, Koscielski I, et al. Comparative analysis of YAP/TEAD inhibitors in 2D and 3D cultures of primary human hepatocytes reveals a novel non-canonical mechanism of CYP induction. *Biochem Pharmacol.* 2023
- Pilling D, Karhadkar TR, Gomer RH. High-Fat Diet–Induced Adipose Tissue and Liver Inflammation and Steatosis in Mice Are Reduced by Inhibiting Sialidases. *Am J Pathol* 2021
- Rinella ME, Lazarus J V., Ratziu V, Francque SM, Sanyal AJ, Kanwal F, et al. A multisociety Delphi consensus statement on new fatty liver disease nomenclature. *J Hepatol* 2023a
- Rinella ME, Lazarus J V., Ratziu V, Francque SM, Sanyal AJ, Kanwal F, et al. A multisociety Delphi consensus statement on new fatty liver disease nomenclature. *Ann Hepatol.* 2024
- Rinella ME, Neuschwander-Tetri BA, Siddiqui MS, Abdelmalek MF, Caldwell S, Barb D, et al. AASLD Practice Guidance on the clinical assessment and management of nonalcoholic fatty liver disease. *Hepatology* 2023b

31st Jan 2024

Manuscript Number: MSB-2023-11999R

Title: Estrogen receptor activation remodels TEAD1 gene expression to alleviate hepatic steatosis

Dear Claudia,

Thank you for sending us your revised manuscript. We have now heard back from two of the three reviewers who agreed to evaluate your revised study. Unfortunately, reviewer #2 did not respond to our requests to evaluate the revised manuscript. As you will see below, reviewer #3 is satisfied with the performed revisions and supports publication. However, reviewer #1 still brings up some concerns, mainly regarding the level of mechanistic insight provided by the study. As this reviewer does not make specific recommendations in terms of what that they would like to see addressed further and given that reviewer #3 is supportive of publication, we think that it seems reasonable to proceed with publishing the study, pending some text modifications to address the remaining concerns of reviewer #1. Specifically, we would ask you to perform a minor revision and make sure that the text reflects potential limitations, alternative mechanistic explanations etc. as mentioned in the reviewer's comments. Please label the revised text clearly so that we can easily access the edits. We would also ask you to address some editorial issues listed below.

- Our data editors have indicated that the following needs to be corrected in the figure legends:

-- The legend for figure EV 1e is not provided, please correct.

-- Please indicate the statistical test used for data analysis in the legends of figures EV 1e; EV 2d; EV 4e; EV 5b.

-- The information related to n is missing in the legends of figures 5a, f.

-- The error bar is not defined in the legend of figure EV 5a.

- The funding information provided in the manuscript text (Acknowledgements) should match the information entered in the online submission system. The following information is missing from the submission system: KI-SRP Diabetes (2023, CK), Lillian Sagen & Curt Ericsson research foundation (2021-00427, CK), Gösta Milton's research foundation (2021-00527, CK); Chinese Scholarship Council (201700260271; KG, CK); the Swedish National Infrastructure for Computing (SNIC) at UPPMAX (storage: 2020/15-225, 2021/23-691; compute: 2020/16-291, 2021/22-860) and National Microscopy Infrastructure, NMI (VR-RFI 2016-00968).

- The References should be formatted according to the Molecular Systems Biology reference style (i.e., ordered alphabetically and listing the first 10 authors followed by et al).

- Please remove the 'Authors Contributions' from the manuscript. The 'Author Contributions' section is replaced by the CRediT contributor roles taxonomy to specify the contributions of each author in the journal submission system. Please use the free text box in the 'author information' section of the online submission system to provide more detailed descriptions if needed (e.g., 'X provided intracellular Ca⁺⁺ measurements in fig Y').

- Our data integrity analyst has detected an image re-use between Figure 1B and Figure EV1D. We would ask you to indicate the image reuse in the respective figure legends for transparency.

Please resubmit your revised manuscript online, with a covering letter listing amendments and responses to each point raised by the referees. Please resubmit the paper ****within one month**** and ideally as soon as possible. If we do not receive the revised manuscript within this time period, the file might be closed and any subsequent resubmission would be treated as a new manuscript. Please use the Manuscript Number (above) in all correspondence.

Click on the link below to submit your revised paper.

Kind regards,

Maria

Maria Polychronidou, PhD

If you do choose to resubmit, please click on the link below to submit the revision online before 29th Feb 2024.

IMPORTANT: Please note that corresponding authors are required to supply an ORCID ID for their name upon submission of a revised manuscript (EMBO Press signed a joint statement to encourage ORCID adoption). (<https://www.embopress.org/page/journal/17444292/authorguide#editorialprocess>)
Currently, our records indicate that the ORCID for your account is 0000-0002-8047-0058.

Please click the link below to modify this ORCID:
Link Not Available

*** PLEASE NOTE *** As part of the EMBO Press transparent editorial process initiative (see our Editorial at <https://dx.doi.org/10.1038/msb.2010.72> , Molecular Systems Biology will publish online a Review Process File to accompany accepted manuscripts. When preparing your letter of response, please be aware that in the event of acceptance, your cover letter/point-by-point document will be included as part of this File, which will be available to the scientific community. More information about this initiative is available in our Instructions to Authors. If you have any questions about this initiative, please contact the editorial office (msb@embo.org).

Reviewer #1:

In the opinion of this reviewer, the revised manuscript unfortunately does not provide much more insights into relevance of the findings towards sex differences in sensitivity to MAFLD and key aspects of the proposed mechanisms have not been convincingly further defined such as the specific liver cell(s) involved in the proposed ER/TEAD1 connection, the direct regulation through ER DNA binding or whether ER activation actually leads to induction of Tead1 in hepatocytes. Concerns about several experiments also remain (e.g. RNAseq on primary human hepatocytes from a single donor, ChIPseq consistency across replicates not taken into account)

Reviewer #3:

In my view, the authors have satisfactorily updated their manuscript according to the comments raised by the reviewers. I therefore recommend accepting this paper for publication.

We thank the reviewers for their insightful evaluations and feedback on our manuscript. In response to reviewer 1's comments, we have verbally addressed the concerns by incorporating additional statements into the discussion, limitation, and materials and methods sections. To provide clarity, we have categorized the reviewer's comments into separate points. The original comments from the reviewers are presented in black and our responses are indicated in blue. Newly added sentences are highlighted in dark green in the manuscript, and revisions made in previous iterations are still marked in red in the manuscript.

Reviewer #1:

Comment 1:

In the opinion of this reviewer, the revised manuscript unfortunately does not provide much more insights into relevance of the findings towards sex differences in sensitivity to MAFLD.

We added an additional statement under the limitations section:

“This study primarily examined the effect of ER agonist treatment on male mice, given that female mice fed with a high-fat diet showed protection against hepatic steatosis. Although our findings indicate that the ER-sensitive genes identified in males are also controlled by estrogen signaling in female mice, further assessment is required to determine whether these genes partially account for the sex disparity observed in MASLD.”

Comment 2:

Key aspects of the proposed mechanisms have not been convincingly further defined such as the specific liver cell(s) involved in the proposed ER/TEAD1 connection, the direct regulation through ER DNA binding or whether ER activation actually leads to induction of Tead1 in hepatocytes.

We added an additional statement in the discussion:

“While the direct regulation of the TEAD1 gene by ERs in hepatocytes is plausible, our data cannot preclude the involvement of various hepatic cell types and secondary signals, which requires further exploration.”

Comment 3:

Concerns about several experiments also remain (e.g. RNAseq on primary human hepatocytes from a single donor)

We kindly want to point out that we addressed this concern in the previous revision by including a dedicated section. We believe this adequacy covers the reviewer's comment:

“Furthermore, inter-individual variability, including demographic, environmental and genetic factors can impact outcomes when working with human primary cells. However, previously we did not observe major differences in the molecular effects of YAP/TEAD inhibitors (Oliva-Vilarnau et al, 2023), despite MASH phenotypes slightly varied when growing spheroids derived from different donors. While these findings argue against major differences of the molecular networks underlying YAP/TEAD inhibition, a modulating role of various individual factors cannot be excluded.”

Comment 4:

ChIPseq consistency across replicates not taken into account

We added a sentence into the Materials and Methods section of the manuscript:

“Of note, ChIP-seq experiments were performed in two batches using H3K27ac antibodies from different lots, which could introduce batch-driven variation.”

Reviewer #3:

In my view, the authors have satisfactorily updated their manuscript according to the comments raised by the reviewers. I therefore recommend accepting this paper for publication.

Thank you.

13th Feb 2024

Manuscript number: MSB-2023-11999RR

Title: Estrogen receptor activation remodels TEAD1 gene expression to alleviate hepatic steatosis

Dear Claudia,

Thank you again for sending us your revised manuscript. We are now satisfied with the modifications made and I am pleased to inform you that your paper has been accepted for publication.

Kind regards,

Maria

Maria Polychronidou, PhD
Senior Editor
Molecular Systems Biology
